# Optical nanomanipulation on solid substrates via optothermally-gated photon nudging

Jingang Li [1], Yaoran Liu[1,2,3], Linhan Lin[1,2], Mingsong Wang[2], Taizhi Jiang[4], Jianhe Guo[1], Hongru Ding [2], Pavana Siddhartha Kollipara[2], Yuji Inoue[2], Donglei Fan [1,2], Brian A. Korgel[4] & Yuebing Zheng [1,2]*

Constructing colloidal particles into functional nanostructures, materials, and devices is a promising yet challenging direction. Many optical techniques have been developed to trap, manipulate, assemble, and print colloidal particles from aqueous solutions into desired configurations on solid substrates. However, these techniques operated in liquid environments generally suffer from pattern collapses, Brownian motion, and challenges that come with reconfigurable assembly. Here, we develop an all-optical technique, termed optothermally-gated photon nudging (OPN), for the versatile manipulation and dynamic patterning of a variety of colloidal particles on a solid substrate at nanoscale accuracy. OPN takes advantage of a thin surfactant layer to optothermally modulate the particle-substrate interaction, which enables the manipulation of colloidal particles on solid substrates with optical scattering force. Along with in situ optical spectroscopy, our non-invasive and contactless nanomanipulation technique will find various applications in nanofabrication, nanophotonics, nanoelectronics, and colloidal sciences.

[1] Materials Science and Engineering Program and Texas Materials Institute, The University of Texas at Austin, Austin, TX, USA. [2] Walker Department of Mechanical Engineering, The University of Texas at Austin, Austin, TX, USA. [3] Department of Electrical and Computer Engineering, The University of Texas at Austin, Austin, TX, USA. [4] McKetta Department of Chemical Engineering, The University of Texas at Austin, Austin, TX, USA. *email: zheng@austin.utexas.edu

The state-of-the-art chemical synthesis techniques permit the production of colloidal particles with precisely tunable sizes and shapes, tailorable compositions and unique properties[1–5]. To build these colloidal particles into functional devices, the particles need to be assembled into the desired nanostructures and transported from an aqueous solution onto a solid substrate. A number of optical techniques, including optical tweezers, have been invented to trap, manipulate, and assemble colloidal particles in fluidic environments at single-particle resolution[6–10]. However, the desired immobilisation of the optically assembled colloidal structures onto solid substrates is not trivial. Along this line, various optical printing methods have been developed to pattern colloidal particles onto substrates[11], such as optoelectric printing[12], plasmon-enhanced laser printing[13], photochemical printing[14], optothermal printing[15] and bubble printing[16]. Despite their ability to pattern colloidal particles into various configurations, one major drawback for printing particles in liquid environments is that the strong capillary force can change the positions of particles and cause the pattern to collapse[17,18]. Additionally, Brownian motion of nanoparticles in the colloidal suspension can interrupt the manipulation process and limit the printing precision[19,20]. van der Waals interactions are strong enough to be exploited to firmly bond particles on the substrate;[13] however, reconfigurable patterning becomes impossible, which prevents the on-demand construction of active nanoarchitectures.

An alternative strategy to overcome these limitations is to dynamically manipulate nanoparticles on a solid substrate. An atomic force microscope (AFM) can manipulate nanosized particles on a flat substrate with nanometre accuracy[21–24]. Unfortunately, AFM manipulation relies on physically pushing the particle with a sharp AFM tip, which often causes undesired tip and particle deformation as well as particle adhesion to the tip. In comparison to the solid-liquid interfaces, the van der Waals friction at the solid-solid interfaces is much stronger, which makes the manipulation of particles on solid substrates a considerable challenge[25,26]. Thus, the key to the effective manipulation of particles on a solid substrate lies in the modulation of interfacial interactions to reduce the friction forces.

Herein, we report a novel optical nanomanipulator, which provides a non-invasive and contactless strategy to achieve versatile nanomanipulation of colloidal particles and nanowires on a solid substrate through interfacial engineering. In short, a thin surfactant layer is introduced between the particles and the glass substrate, acting as an optothermal gate to modify particle-substrate interfacial interactions. With the optical heating of the particles, the friction of the particle and surfactant is dramatically reduced due to the phase transition of the surfactant layer, allowing the manipulation of particles with optical scattering forces[27,28]. We term our technique optothermally-gated photon nudging (OPN). OPN is capable of dynamic manipulation and reconfigurable patterning of colloidal particles with a wide range of materials, sizes, and shapes on solid substrates. In combination with in situ dark-field optical imaging and spectroscopy, we can visualise the manipulation process with real-time feedback and measure the properties of the particles and their interactions in assemblies.

## Results

### General concept.
The general concept of OPN is illustrated in Fig. 1a. A thin layer of surfactant is deposited between the glass substrate and the randomly dispersed colloidal particles (see Methods for details; also see Supplementary Fig. 1 for experimental setup and the detailed configuration of the sample). For demonstration, we used cetyltrimethylammonium chloride (CTAC) as our thin layer; however, it can be substituted with any other surfactant or polymer with similar photothermal responses, such as poly(methyl methacrylate). The deposited CTAC acts as an optothermal gate to modulate the particle-substrate interface and allows for the manipulation of particles, which is pivotal for OPN. Without optical heating, CTAC forms a thin solid film (Supplementary Fig. 2)[29] and particles adhere to the film with van der Waals forces (Fig. 1b). To release the bond between the film and the particle, we directed a laser beam onto the particle, whose optothermal effects generate an abundance of heat. The maximum temperature reached over 600 K when a 200 nm AuNP was irradiated by a 532 nm laser beam at an optical power of 1 mW (Fig. 1d; see Methods for simulation details). In addition, the temperature of the CTAC layer under the AuNP exceeded ~450 K, which is larger than the first-order phase transition temperature of CTAC at 350–370 K[29]. Under such high temperatures, CTAC surrounding the particle undergoes a localised order-disorder transition and turns into a quasi-liquid phase (Fig. 1c), where the nonpolar layers are melted while the ionic layers remain practically intact[29,30]. This disordered structure significantly eliminates the van der Waals friction between the particle and CTAC layer, opening the optothermal gate for free particle motion. With the optothermal gate open, particles can be nudged and guided smoothly by the laser beam with optical scattering forces (Fig. 1e). Through steering the laser beam or translating the substrate with a motorised stage, particles can be manipulated to any target position. It should be noted that this work is quite different from opto-thermoelectric nanotweezers published in a recent paper[31]. In our OPN, particles are manipulated on solid substrates by optical scattering forces. In contrast, opto-thermoelectric nanotweezers exploit CTAC that is dissolved into a colloidal solution to generate an opto-thermoelectric field to trap charged nanoparticles.

We first present the use of OPN for manipulating gold nanoparticles (AuNPs) as a proof-of-concept demonstration. AuNPs were tracked with in situ dark-field optical imaging due to their strong light scattering properties. Figure 1f demonstrates the real-time manipulation process of a 300 nm AuNP using a 532 nm laser. The power intensity we used was 0.2–2 mW/$\mu$m$^2$, which is ~2 orders of magnitude lower than the typical power intensity in optical tweezers (10-100 mW/$\mu$m$^2$). The AuNP was delivered in-plane over a distance of ~8 $\mu$m in 35 s. Apart from AuNPs, we also demonstrated the nanomanipulation of other materials using OPN, such as silver nanoparticles (AgNPs) and silicon nanoparticles (SiNPs). Particles with a wide range of diameters from 40 nm to several micrometres can be manipulated with an average speed of 0.2–2 $\mu$m/s (Supplementary Movie 1). It should be noted that the speed of the OPN manipulation was limited by the manual operation in this scenario. With automatic digital operation and feedback controls, the manipulation speed could be further improved. Interestingly, the CTAC layer remains functional as a vital component of OPN after the particle translation (Supplementary Fig. 3), which allows the particle to be steered back to its original position along the same path (see Supplementary Movie 1 for the manipulation of an 80 nm AuNP).

### Characterisations of OPN manipulation.
In the following section, we discuss the underlying physical mechanisms and analyse the forces involved in OPN in detail. First, we designed and conducted a series of control experiments to understand the role of optical heating and scattering force, as summarised in Supplementary Table 1. To decouple the optical heating and scattering, we applied a thermoplasmonic substrate as the heat source (see Supplementary Fig. 4 for the structure and Methods for

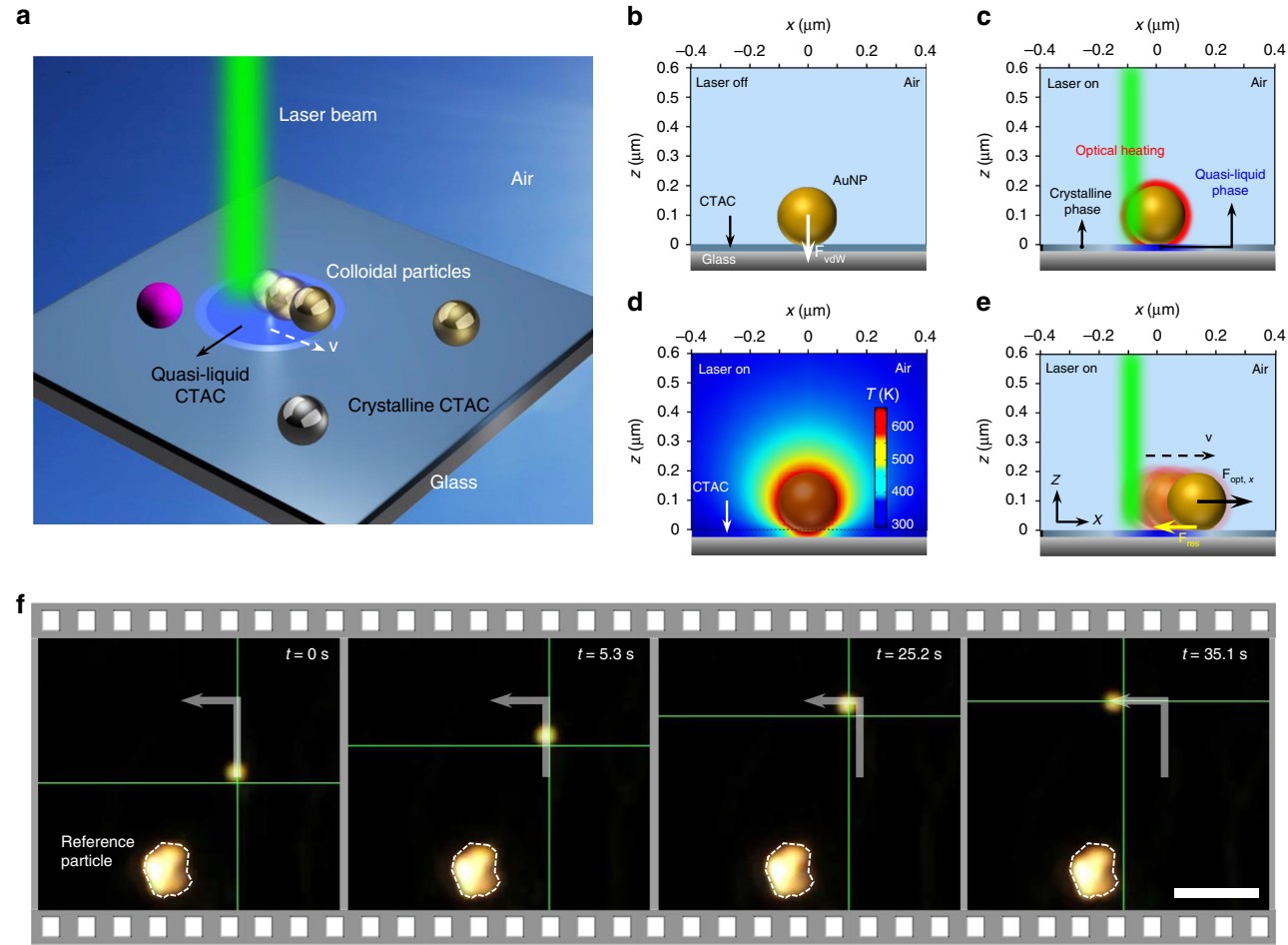

**Fig. 1 General concept of OPN. a** Schematic illustration of OPN on a solid substrate. **b** A 200 nm AuNP placed on and bonded with the CTAC layer by van der Waals force $\mathbf{F}_{vdW}$ without optical heating. **c** The optical heating under laser illumination induces a localised phase transition in the surrounding CTAC layer. CTAC turns into a quasi-liquid phase and releases the bond with AuNP. **d** The simulated temperature distribution around a 200 nm AuNP. Incident power: 1 mW; laser beam size: 0.8 μm. **e** AuNP moves against the laser beam with an in-plane optical force $\mathbf{F}_{opt,\,x}$ and a resistant force $\mathbf{F}_{res}$. In **a–e**, the schematic configuration is flipped upside down for better visualisation. **f** Sequential dark-field optical images showing real-time manipulation of a 300 nm AuNP. The green crosshair indicates the position of the laser beam. The white arrow depicts the path. Scale bar: 5 μm.

fabrication procedure)[32]. In addition, we selected polystyrene (PS) and titanium dioxide ($TiO_2$) nanoparticles for the control experiments because of their negligible optical absorption and distinct scattering properties (Supplementary Fig. 5)[33,34]. The results revealed that OPN simultaneously exploits optical heating to open the optothermal gate and radiation-pressure forces to drive the particles (see Supplementary Note 1 and Supplementary Movies 1 and 2). Moreover, the option to introduce an external heat source to trigger the manipulation of particles makes OPN a generalised platform for a wide range of materials that interact strongly with light, such as metal (e.g., aluminum)[35], semiconductor (e.g., germanium and gallium arsenide)[36,37], and inorganic perovskite (e.g., barium titanate)[38]. We further conducted experiments to exclude the thermal expansion force and electrostatic force as the primary driving forces in OPN (see Supplementary Note 2, Supplementary Fig. 12, and Supplementary Movies 3 and 4).

Next, we quantitatively analysed the OPN manipulation by measuring the velocities of 200 nm and 300 nm AuNPs under a directed laser with a fast CCD (Fig. 2a; also see Methods for details of measurement). The measured particle velocity is the result of the balance between the optical driving force and the resistant force by surfactant. As an example, the measured data for a 300 nm AuNP at an optical power of 1.40 mW is shown in

Fig. 2b. When the laser was on, the AuNP immediately gained an in-plane speed and moved in a direction against the laser beam. Since the laser beam was focused slightly offset from the particle center along the X-axis, the AuNP had a much larger shift in the X-direction than that in the Y-direction, which is consistent with our photon nudging hypothesis. As we raised the laser power, the AuNPs increased their speeds and shifted farther from their original positions (Supplementary Movie 5; also see Supplementary Fig. 6 for the detailed data). For both 200 and 300 nm AuNPs, the measured maximum velocities increased when the optical power was raised from 0.27 to 1.40 mW (Fig. 2c). This relationship is reasonable, considering that the optical force scales linearly with the laser power, which further confirms the optical force is the primary driving force in OPN.

Furthermore, we adopted a simplified physical model to understand the nanomanipulation process. Since the particles were manipulated in the X-Y plane, only the in-plane optical and resistant forces were considered. We applied the finite-difference time-domain (FDTD) simulations to calculate the optical scattering forces in the X-direction with varying distances between AuNPs and the laser beam (Fig. 2f, also see Methods for simulation details). It should be noted that for both X- and Y-polarisations, the laser beam will always repel the AuNP, which allows OPN to maneuver particles in all directions without the

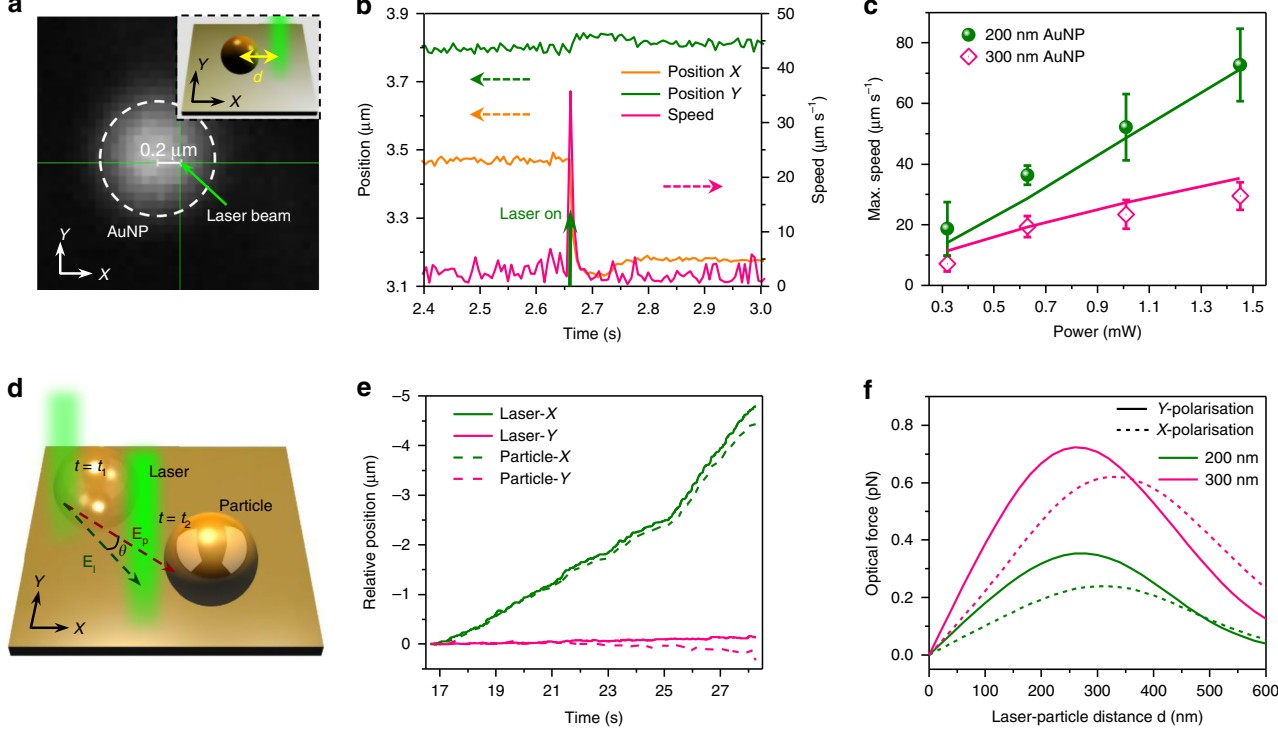

**Fig. 2 Characterisation and modelling of the OPN manipulation process. a** Optical image and (inset) the schematic illustration showing the measurement of particle velocities. The distance between the centre of the laser beam and the centre of the particle was set to 200 nm for all measurements. **b** The measured X position, Y position, and speed v of a 300 nm AuNP under the laser irradiation of 1.40 mW as a function of time t. The solid green arrow at t = ~2.7 s indicates the instant when the laser is turned on. **c** Measured maximum speed of 200 nm and 300 nm AuNPs as a function of incident power. The solid lines show the corresponding modelled data. **d** Schematic of the comparison of the laser movement vector $\mathbf{E}_l$ and the particle movement vector $\mathbf{E}_p$ at two successive frames (t = t₁, t₂). θ is denoted as the angle between $\mathbf{E}_l$ and $\mathbf{E}_p$. **e** Examples of the recorded trajectories of the laser beam and the particle during the manipulation (see Supplementary Movie 1 for the manipulation of the 80 nm AuNP). **f** The calculated optical scattering forces of 200 nm (olive) and 300 nm (pink) AuNPs as a function of laser-particle distance. Incident power: 1.4 mW. Source data are provided as a Source Data file.

need to control the laser polarisation. In our case, the particles are partially immersed into the CTAC film (Supplementary Fig. 7), for which the resistant force can be evaluated according to:[39]

$$\mathbf{F}_{res} = 6\pi\eta R f_d \mathbf{v} \qquad (1)$$

where η is the viscosity of CTAC in its quasi-liquid phase, R is the particle radius, **v** is the velocity of the particle, and $f_d$ is a dimensionless drag coefficient which is dependent on the viscosity of the fluids[40]. The detailed calculation of the resistant force can be found in Supplementary Note 3 (also see Supplementary Figs. 13 and 14). The trajectory of the AuNP can be modelled with:

$$m\ddot{\mathbf{x}} = \mathbf{F}_{res}(\mathbf{v}) + \mathbf{F}_{opt}(\mathbf{x}) \qquad (2)$$

where m is the mass of the particle, **x** is the position of the particle, and $\mathbf{F}_{opt}(\mathbf{x})$ is the total optical forces calculated by the FDTD. We applied MATLAB to numerically solve the motion of particles in OPN with the same time step as the fast CCD. The calculated maximum velocity values for 200 nm and 300 nm AuNPs under different laser powers are shown in Fig. 2c. The results match well with our measurements, which further confirms our proposed mechanisms.

We further characterise the manipulation efficiency by analysing the video recordings of the particle movement. During the manipulation process, the trajectories of the particles and the laser beam almost overlap (Fig. 2e and Supplementary Fig. 8), which shows that the particles can be efficiently manipulated along the laser direction. To quantify the manipulation efficiency, we examined the difference between the laser movement vector $\mathbf{E}_l$

and the particle movement vector $\mathbf{E}_p$, as sketched in Fig. 2d. The accuracy of the particle movement can be characterised by the dot product of unit vectors along $\mathbf{E}_l$ and $\mathbf{E}_p$. We define a manipulation efficiency Q as the average cosθ over a full manipulation trajectory:

$$Q = \langle\cos\theta\rangle = \left\langle \frac{\mathbf{E}_l \cdot \mathbf{E}_p}{|\mathbf{E}_l||\mathbf{E}_p|} \right\rangle \qquad (3)$$

where θ is the angle between $\mathbf{E}_l$ and $\mathbf{E}_p$. The calculated Q ranges from ~0.6 to 0.8 for the recorded videos (Supplementary Fig. 8), indicating high-efficient manipulation of all kinds of colloidal particles.

**Patterning accuracy**. To assess OPN as a nanomanufacturing tool for arbitrary and precise construction of colloidal structures, we explored the patterning accuracy of OPN in both 1D and 2D cases. As a preliminary demonstration, we used OPN to assemble seven randomly dispersed SiNPs with a diameter of 500 nm into a straight line (Fig. 3a, b). The dark-field image shows a well-arranged particle chain after the patterning procedure (Fig. 3c). Furthermore, we took the SEM image of the particle assembly to precisely determine their positions and their deviations from the target line (Fig. 3d). The position error, the distance between the particle centre and the line (inset in Fig. 3d), was analysed for individual SiNPs and plotted in Fig. 3e. A position accuracy of ~80 nm was achieved, as indicated by the shaded area.

In the second example, we manipulated nine SiNPs using OPN to assemble a 3 × 3 2D array (Fig. 3f). The optical image and corresponding SEM image of the SiNP array are shown in Fig. 3g, h.

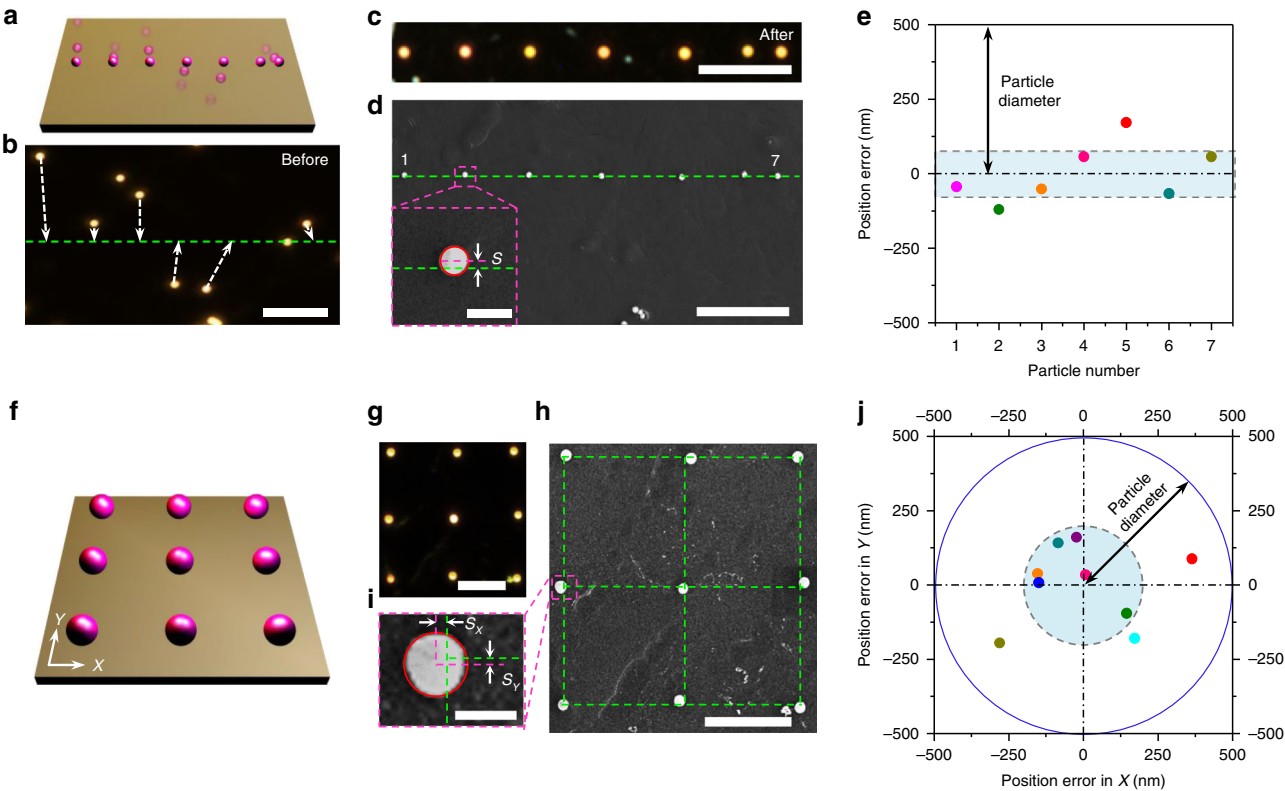

**Fig. 3 Patterning accuracy of OPN. a** Schematic illustration of the 1D assembly of seven 500 nm SiNPs. **b** Dark-field image of SiNPs before patterning. The white dashed arrows indicate the target positions of each SiNPs. **c** Optical image of SiNPs after 1D assembly. **d** SEM image of SiNPs after 1D assembly. The inset defines the method to determine the position error $S$ for individual SiNPs. **e** Position errors for each SiNPs in the line. The shaded area shows the average deviation from the target line, which is ~80 nm. **f**, **g** Schematic illustration and dark-field image of a 2D assembly of nine 500 nm SiNPs into a 3 × 3 array. **h** SEM image of the 2D assembly. **i** The position error in $X$ ($S_X$) and $Y$ ($S_Y$) for individual SiNPs. **j** Position errors in $X$ and $Y$ for each SiNPs in the 2D array. The shaded area indicates the average deviation from the target positions, which is ~200 nm. Scale bars: **b**–**d** 10 μm; inset in **d**, 1 μm; **g**, **h**, 5 μm; **i**, 500 nm.

Similar to the 1D case, we evaluated the position error in both the $X$ and $Y$ directions from the SEM image (Fig. 3i). As depicted in Fig. 3j, all manipulated particles are located close to their target positions with an average deviation of ~200 nm, which is less than half the diameter of the SiNPs (see Supplementary Fig. 9 and Supplementary Table 2 for the detailed information). The ability to achieve colloidal patterning at nanoscale accuracy enables OPN to be used for the precise fabrication of nanostructures with colloidal particles. It should be noted that the current positional accuracy is primarily limited by the diffraction barrier in optical microscopy. Additionally, during the manipulation experiments, we only relied on visual estimations to determine the positions of the nanoparticles in the optical images. Thus, we envision that the particle-patterning accuracy can be further improved with advanced imaging, analysis and tracking of particles with higher precision. For instance, we have achieved an improved position accuracy of ~20 nm by using the imaging software to define a target line along which the particles will be aligned (Supplementary Note 4 and Supplementary Fig. 15). Moreover, OPN is capable of on-demand patterning of colloidal particles into more complex configurations (Supplementary Fig. 10). It is also worth noting that the CTAC layer is removable without altering the particle positions after OPN patterning, which will be discussed in detail later.

**Reconfigurable patterning**. Since the nanomanipulation is performed on a solid substrate, OPN allows for the dynamic transportation of particles to new sites, enabling active assembly of colloidal structures. Reconfigurable patterning of four 300 nm AuNPs is shown in Fig. 4a. The randomly dispersed AuNPs were

first assembled into an L-shaped structure. By moving the top particle down to the right side, we transformed the L-shaped pattern into a square. Then, the particle at the upper left corner of the square was translated to the top right, forming a mirrored L-shaped pattern. Finally, four AuNPs were assembled into a straight line by delivering the particle on the left to the bottom.

Apart from maneuvering spherical colloidal particles, we also achieved dynamic manipulation of metallic nanowires through OPN. Gold nanowires (AuNWs) with a diameter of 160 nm and a length of 3 μm were used in this experiment. Since gold has high thermal conductivity, the CTAC optothermal gate can be triggered with the laser beam directed at any location along the AuNWs. By focusing the laser at one end of the nanowire while steadily moving the laser tangentially, the nanowire can be rotated about the opposite end, which remains fixed. A single AuNW can be rotated over ~180 degrees in a counter-clockwise direction within 32 s (Fig. 4b; also see Supplementary Movie 6). AuNW translation is also possible by directing the laser at the centre of the nanowire, which allows us to transport a AuNW with a fixed orientation. Naturally, due to the inability to place the laser beam strictly at the nanowire centre, the AuNW will rotate slightly during this translation process. This phenomenon can be easily avoided by implementing a feedback loop to rotate the AuNW in the opposite direction to its original orientation. As a demonstration, we transported a AuNW translationally over a distance of ~5 μm within 23 s (Fig. 4c; also see Supplementary Movie 6).

Furthermore, we were able to achieve the reconfigurable patterning of hybrid nanostructures comprised of a metallic nanowire and dielectric nanoparticles. Two SiNPs and one

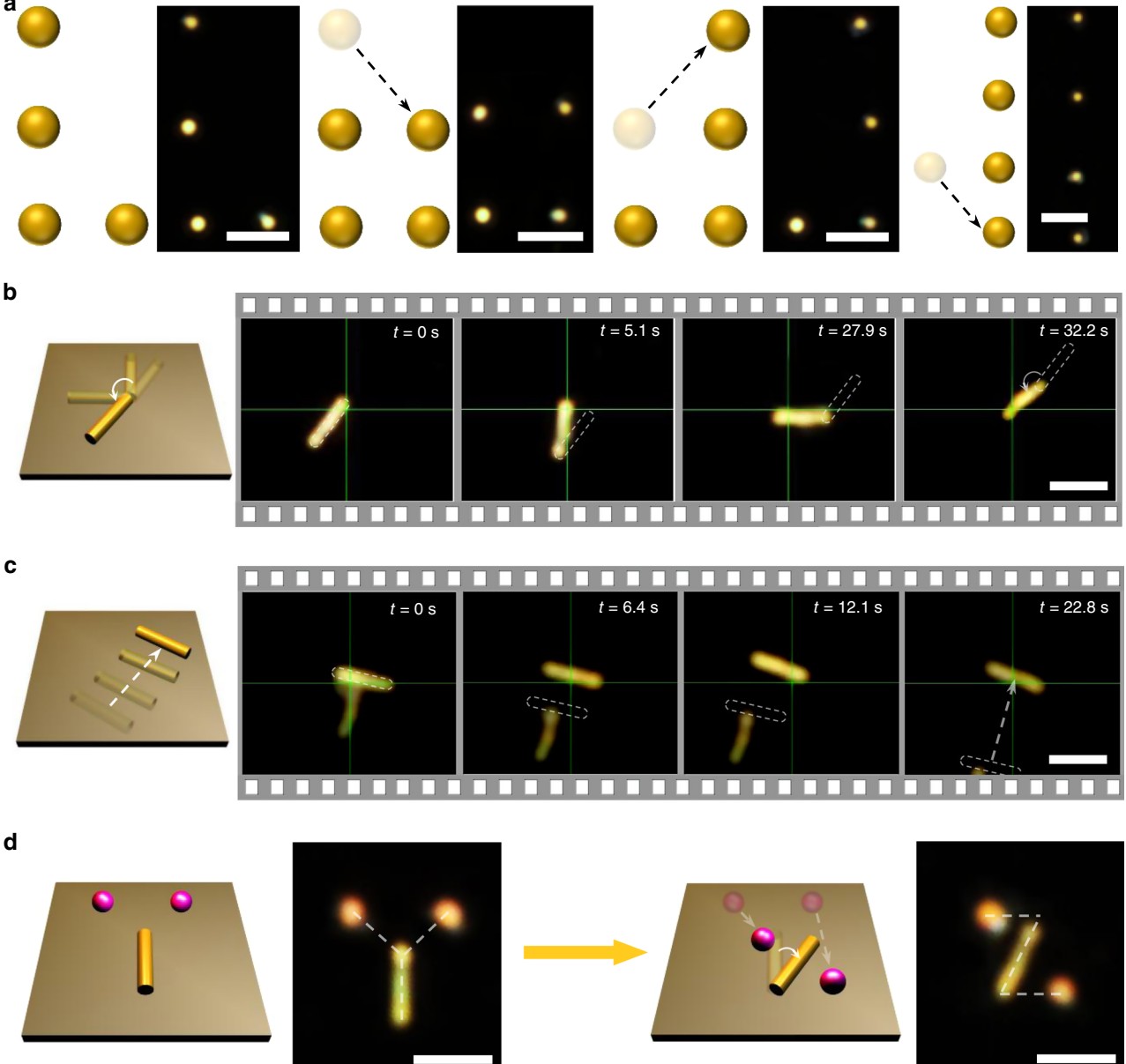

**Fig. 4 Reconfigurable patterning of particles and nanowires. a** Reconfigurable patterning of four 300 nm AuNPs. Four particles were arranged into L-shape, a square, mirrored L-shape, and a straight line, sequentially. The dashed arrows show the reconfigurable patterning sequence. **b, c** Schematic illustrations and successive optical images showing the real-time (**b**) rotation and (**c**) translation of AuNWs. The green crosshairs mark the positions of the laser beam. The dashed rectangular outline indicates the original positions of nanowires. **d** Schematic illustration and optical images showing reconfigurable patterning of metal-dielectric hybrid nanostructures. Two 500 nm SiNPs and one AuNW were patterned into "Y" and "Z", sequentially. Scale bars: **a**, 5 μm; **b–d** 3 μm.

AuNW were first patterned into a Y-shaped structure. By rotating the AuNW and by moving SiNPs to new sites, we deliberately transformed the structure into a Z-shaped pattern (Fig. 4d). The ability to dynamically manipulate nanowires and metal-dielectric nanostructures shows the potential of OPN for the assembly of functional components and devices. It is worth noting that OPN can also be applied to manipulate other anisotropic nanoobjects, such as gold nanorods and gold nanotriangles[41].

**In situ optical spectroscopy**. We further apply in situ spectroscopy to study the spectral response of colloidal nanostructures. The minimal backward scattering from the CTAC layer permits the detection of the intrinsic scattering spectra from the particles. The in situ optical spectroscopy is capable of distinguishing colloidal particles with different sizes by comparing their scattering peak positions during the manipulation process (Supplementary Note 6 and Supplementary Fig. 16). We first measured the scattering spectra of a 100 nm AuNP before and after the manipulation to show that OPN can manipulate nanoparticles without damaging their optical properties (Supplementary Fig. 11). This non-invasive operation is highly desired and advantageous in the fabrication of functional components and devices such as reconfigurable optical nanocircuits and active plasmonic waveguides. We further showed that the CTAC layer can be readily removed without destroying the existing particle patterns by simply soaking the sample in water or isopropyl alcohol (IPA) for ~2 min. As shown in Fig. 5a, b, the positions of the colloidal particles remained the same after the removal of CTAC.

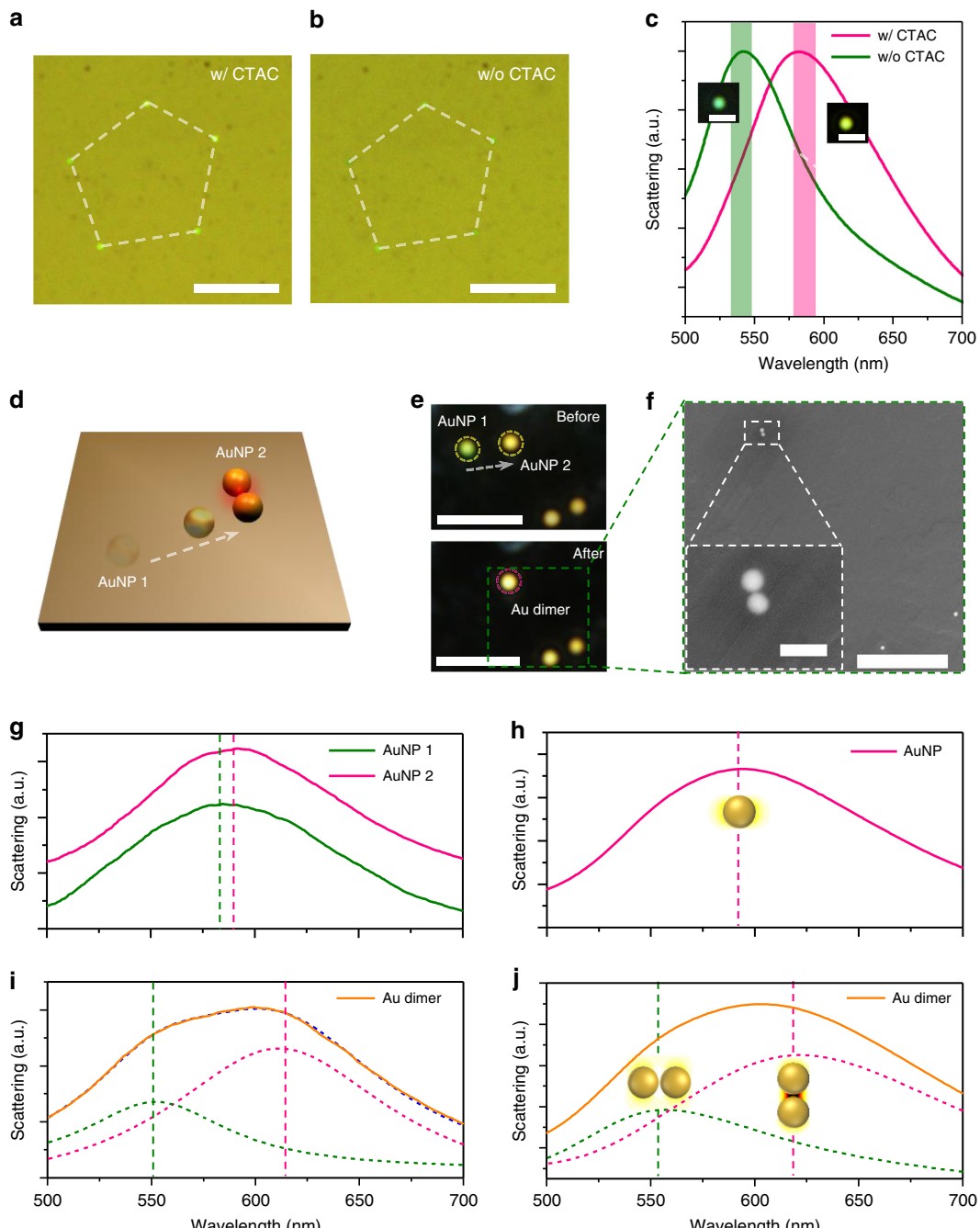

**Fig. 5 In situ optical spectroscopy. a**, **b** Optical images of a pentagon pattern composed of five 100 nm AuNPs **a** before and **b** after the removal of CTAC. Two identical white dashed pentagons are added to help indicate the positions of AuNPs. **c** The scattering spectra of 100 nm AuNPs measured before and after the removal of CTAC. The shaded area indicates the standard deviation of the peak position. The insets show the corresponding dark-field images. **d** Schematic illustration of the assembly of Au dimer with two 100 nm AuNPs. **e** Dark-field images of the AuNP before and after the dimer assembly. **f** SEM image of the Au dimer. **g** Scattering spectra of the two AuNPs before assembly. **h** The simulated scattering spectra of a 100 nm AuNP. **i** Scattering spectra of the Au dimer. The pink and olive dashed curves represent the longitudinal and transverse plasmon modes, respectively. **j** The simulated scattering spectrum of the AuNP dimer. The insets in **h**, **j** show the corresponding electric field enhancement profiles. Scale bars: **a**, **b** 10 μm; inset in **c** 2 μm; **e** 5 μm; **f** 2 μm; inset in **f** 200 nm.

Meanwhile, the measured scattering peak of 100 nm AuNPs showed an obvious blueshift from ~580 to ~545 nm (Fig. 5c). This blueshift revealed a refractive-index change in the particle surrounding, which confirmed the successful removal of CTAC (the refractive index of CTAC is 1.38). The ability to remove CTAC after OPN manipulation can avoid any undesirable effects of CTAC in some of applications of the patterned particles, including chemical and biological sensing.

Furthermore, we explored OPN's potential to assemble and characterise colloidal structures with near-field coupling. One 100 nm AuNP was delivered to the vicinity of the other 100 nm AuNP by OPN, as depicted in Fig. 5d, e. The assembled structure was confirmed by the SEM image, from which a clear dimer with a gap of ~15 nm can be observed (Fig. 5f). Before assembly, the single AuNPs showed a localised surface plasmon (LSP) peak at ~588 nm (Fig. 5g). The small difference in the LSP peaks of these

two AuNPs resulted from the slight variations in the particle sizes, as shown in Fig. 5f. The single scattering peak split into two peaks at ~550 and ~614 nm (Fig. 5i), which unequivocally revealed the near-field coupling between two AuNPs. We simulated the scattering spectra and electric field enhancement profiles of a single 100 nm AuNP and a 100 nm AuNP dimer with a gap of 15 nm (Fig. 5h, j). The AuNP dimer exhibited a longitudinal mode at ~618 nm and a transverse mode at ~553 nm, which signify a redshift and a blueshift, respectively, contrasting from the original dipole mode at ~590 nm for the single 100 nm AuNP[42]. The simulation results matched very well with the experimental spectra shown in Fig. 5g, i. It is worth noting that, although the sub-wavelength interparticle gap cannot be distinguished in optical images due to the diffraction limit, OPN can reliably fabricate Au dimers with any desired gap by taking advantage of the in situ optical spectroscopy (Supplementary Note 6 and Supplementary Fig. 17). The ability to control the near-field coupling of nanoparticles allows us to fabricate functional colloidal devices for a variety of nanophotonic applications, such as chiral metamolecules[43]. With simultaneous reconfigurable nanofabrication on a solid substrate and in situ optical characterisations, OPN will provide a powerful platform to design active optical devices and study the coupling between colloidal structures. Furthermore, OPN can be applied in a vacuum or an inert gas environment to assemble water-soluble and air-unstable nanoparticles (e.g., halide perovskite nanoparticles[44,45]) and explore the light-matter interactions in combination with other vacuum-based analytical tools, such as scanning transmission electron microscopy and cathodoluminescence spectroscopy.

## Discussion

Through coordinating optical heating and radiation-pressure forces, we developed an OPN technique for nanomanipulation and patterning of colloidal particles and nanowires on a solid substrate. OPN represents a milestone in pushing the working conditions of optical tweezers from fluidic to solid phases. As a general solid-phase optical technique, OPN is applicable to a wide range of metal, semiconductor, metal oxide and dielectric nanoparticles with varying sizes and shapes. By improved heat management and proper choice of working wavelengths (Supplementary Note 7 and Supplementary Fig. 18), OPN is readily extended to manipulate many other particles that exhibit strong ultraviolet or near-infrared responses such as aluminium nanoparticles[35] and titanium nitride nanoparticles[46]. OPN can dynamically pattern colloidal particles into any desired configurations. However, it remains challenging to achieve sub-20 nm position accuracy and orientational control of anisotropic nanoparticles due to the optical diffraction limit.

Future efforts can be made to further enhance the strengths of OPN. One can optimise the optics to achieve a more efficient operation. For instance, oblique incidence of the laser can take advantage of photon momentum along the direction of beam propagation, which could enhance both amplitude and directional control of the driving forces. While OPN offers the opportunity to manipulate colloids at single-particle resolution, it suffers from relatively low patterning throughput, which is primarily limited by its serial and manual control. The implementation of a light spatial modulator with a digital feedback control will open up the possibilities for automatic and parallel manipulation to significantly boost the production output.

Along with the development of OPN, we have advanced the fundamental understanding and dynamic control of particle-substrate and light-particle interactions. With the in situ optical spectroscopy, OPN holds the potential to dynamically assemble colloidal matters and to explore the mechanical, electronic and optical couplings between colloidal particles at the nanoscale.

With its simple optics, non-invasive operation and versatile capabilities of colloidal assembly, OPN will find a wide range of applications in nanophotonics, nanoelectronics, materials science, and colloidal sciences.

## Methods

**Materials preparation**. CTAC was purchased from Chem-Impex. SDS (20%) solution was purchased from Fisher Bioreagents. In all, 40 nm, 80 nm, 200 nm, 300 nm, 400 nm AuNPs and TiO$_2$ nanoparticles (anatase phase) were purchased from Sigma-Aldrich. In all, 80 nm AuNPs and 110 nm AgNPs were purchased from nanoComposix. In all, 1 and 1.5 μm AuNPs were purchased from Nanopartz. 500 nm polystyrene colloids were bought from Bangs Laboratories. SiNPs were prepared using previously reported synthesis protocols[47]. AuNWs were synthesized using the reported method[48]. 0.5 M CTAC solution in isopropyl alcohol (IPA) was spin coated on to a glass substrate to form a thin layer of CTAC solid film after IPA evaporation. Nanoparticles and nanowires diluted in ethanol were spin coated on CTAC film for manipulation experiments. The SDS layer was obtained by directly spin coating the purchased SDS solution onto the glass and let it dry at room temperature. The thermoplasmonic substrate was fabricated by a two-step process. First, a 4.5 nm Au film was deposited on a glass substrate with thermal deposition (Denton thermal evaporator) at a base pressure below $1 \times 10^{-5}$ Torr. Then, the Au film was thermally annealed at 550 °C for 2 h.

**Optical setup and in situ spectroscopy**. A Nikon inverted microscope (Nikon Ti-E) with a ×100 oil objective (Nikon, NA 0.5–1.3) and a motorised stage was used for the manipulation experiments. A 532 nm laser (Coherent, Genesis MX STM-1 W) was expanded with a 5× beam expander (Thorlabs, GBE05-A) and directed to the microscope. An oil condenser (NA 1.20–1.43) was used to focus the white incident light onto the sample from the top. A colour charge-coupled device (CCD) camera (Nikon) and a fast monochromic CCD camera (Andor) were used to record optical images and track particles, respectively. The scattering signal from the nanoparticles was directed to a two-dimensional detector in a spectrometer (Andor) with a 500 nm grating. Background spectra were recorded and subtracted to obtain the scattering signal of the particles. The spectra were finally normalised with the light source spectra.

**Measurement of velocity**. The laser beam axis was first set at 200 nm offset from the particle centre in X-direction for both sizes of AuNPs and all incident powers. Then, the laser was turned on and the target AuNPs were tracked by the fast CCD with the minimum timestep of 4 ms. The recorded position, speed, and acceleration were analysed in software (Nikon) and directly exported. The velocity in each measurement was defined as the maximum velocity value at the instant when the laser was turned on.

**Characterisations**. SEM images were taken with a FEI Quanta 650 SEM; AFM images were measured with a Park Scientific atomic force microscope. The positions of laser beam and the particle in the supplementary videos were analysed using MATLAB.

**Numerical simulations**. We simulated the electromagnetic field distribution and the absorption cross-section of nanoparticles using finite-difference time-domain method (Lumerical FDTD). The mesh size was defined as 2 nm for nanoparticles. A refractive index of 1 was used for the surrounding medium. The heat density was calculated by $P_{abs} = \frac{I\sigma_{abs}}{V}$, where $\sigma_{abs}$ was the absorption cross-section obtained using FDTD simulations, $I$ was the illumination intensity, and $V$ was the volume of the nanoparticle. By assuming the high thermal conductivity of the gold, we simulated the temperature field profile using 3D finite element method. The outer boundaries were set at room temperature.

## Data availability

The data underlying Fig. 2c are available in the associated source data file. All other data that support the findings of this study are available from the corresponding author upon reasonable request.

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

## Acknowledgements

Y.Z. acknowledges the financial supports of the Army Research Office (W911NF-17-1-0561), the National Aeronautics and Space Administration Early Career Faculty Award (80NSSC17K0520), the National Science Foundation (NSF-CMMI-1761743), and the National Institute of General Medical Sciences of the National Institutes of Health (DP2GM128446). We also thank the Texas Advanced Computing Centre at The University of Texas at Austin (http://www.tacc.utexas.edu) for providing HPC resources that have contributed to the research results reported within this paper.

## Author contributions

J.L. and Y.Z. conceived the idea. J.L. prepared the materials, worked on the experiments and collected the data. Y.L. worked on the numerical simulations. L.L., M.W. and Y.I. assisted with some experiments and discussions. T.J. and B.A.K. synthesized the SiNPs. J.G. and D.F. synthesised the AuNWs. H.D. worked on molecular dynamics simulations. P.S.K. worked on the MATLAB modelling. Y.Z. supervised the project. All authors participated in the discussion of the results and wrote the paper.

## Competing interests

The authors declare no competing interests.
