## [Peer Review File · Nature Communications]

Reviewers' comments:

Reviewer #1 (Remarks to the Author):

This paper presents a novel mechanism for submicron particle manipulation. It combines a phase change of a material by optothermal heating with optical scattering forces. This interesting mechanism should be of interest to researchers in this field.

Overall, this is an interesting, well-written, and comprehensive paper. However, some further revisions can strengthen this paper further:

1. Since the mechanism pushes a particles under manipulation by optical scattering force, the manipulation is inherently unstable. Please include a better characterization of the accuracy of the particle movement. One possibility would be to make a measurement similar to "optical trapping stiffness" that is regularly characterized for optical tweezers. Another suggestion is to determine the difference between the laser movement vector and the particle movement vector at various points along a trajectory, which could be done by analyzing the existing recordings of particle movement.

2. Equation 1 in the paper is Stokes' Law, but this form of the equation assumes an unbounded fluid. This equation needs a correction factor when the particles are close to a surface, as is the case in this paper. In addition, it is not clear how this equation is applied to find the resistance force. Is it used only for the portion of the particle cross-section traveling in the CTAC layer?

3. In Figure 2b, please change the x-axis to a narrower range, so that the change in velocity when the laser is switched on is more apparent. (There is a steep slope to the velocity, but it should be finite.)

Reviewer #2 (Remarks to the Author):

This paper describes a method terms, termed optothermally-gated photon nudging (OPN), for inducing motion of colloidal particles and nanowires on a solid substrate. This solid-phase optical micromanipulation exploits placement of a thin surfactant layer between the particles to be organised and a substrate. This layer acts as an optothermal gate to optically modulate the particle-substrate interaction. The authors claim this breaks "the limitations of conventional optical tweezers to liquid, vapor or vacuum environments".

The particular modulation of interfacial interactions os proposed to lower the friction forces via the thin surfactant layer is introduced between the particles and the glass substrate, The optical heating of the particles, the friction of the particle and surfactant is dramatically reduced due to the phase transition of the surfactant layer, which allows the manipulation of particles with optical scattering forces

The paper has some interest but falls well short of showing the compelling data or advance over previous literature that would warrant publication in Nature Communications. Some points I would raise is support of my conclusions as well as more general points regarding the paper are below:

1) This paper seems to me a variant of their recent Nature Photonics (<https://www.nature.com/articles/s41566-018-0134-3>, not cited!) where a cationic surfactant, cetyltrimethylammonium chloride (CTAC) is added to the nanoparticle colloid. why is this work not properly described nor placed in context?

2) what is the advantage over

- large area evanescent wave large area traps with added single beam tweezers from above? - e.g adding tweezers to a system like Phys. Rev. B 73, 085417 (2006) or

-light induced dielectrophoretic traps (Ming Wu group)

- other optothermal methods, eg. Chen, Hongtao & Gratton, Enrico & Digman, Michelle. (2016).

Self-assisted optothermal trapping of gold nanorods under two-photon excitation. see Methods and Applications in Fluorescence. 4. 10.1088/2050-6120/4/3/035003.

3) equation (1) , line 154, can't be correct..surely we need to use the Faxen correction given the close proximity of the particles to the surface? What in fact is this distance and how is they determined?

4) A position accuracy of ~80 nm was achieved, as indicated by the shaded area seems poor for the SiNPs with a diameter of 500 nm placed into a straight line... ~15% accuracy? This will have to be improved for any reproducible particle organisation

5) fig 3 is really for such a small number of particles.. it would need 2-3 orders of magnitude more to be convincing for real self organisation

6) The CTAC layer is removable without altering the particle positions after OPN patterning, as CTAC might be best to avoid for some applications. This should be in the main text and discussed in more detail

7) Reconfigurable patterning of four 300 nm AuNPs is interesting but hardly enough for proving organisation

8) what is the optimum choice of laser wavelength for optothermal gate?

9) fabrication of active devices and functional components is mentioned. The authors need to explain in some detail what they might be

10) as inherently still relying on optical scattering which is still pN of force, surely this approach is too slow/weak to be of use for larger scale organisation?

Overall, this is really a modest improvement and the data show are not of the level for a Nature Comm . A revised paper would be good for an ACS journal or App Phys Lett

Reviewer #3 (Remarks to the Author):

This manuscript presents an innovative concept that allows a light beam to control and pattern nanoscale particles. Their impact comes from the idea of using a layer of surfactant molecules that can change between solid and fluid phases at locations under light illumination and heat generation. Without light illumination, the nanoparticles are stably anchored on the surface. Under illumination, nanoparticles are pushed to move on a 2D surface due to a combination effect of optical scattering forces and molten surfactant.

This is the most unique part of this work compared to other optical manipulation mechanisms that need to always compete with brownian motion and have difficulty to maintain particle locations once light beams are removed. To understand the true working mechanism behind this interesting phenomenon, the authors have done a nice comparison of different kind of substrate and nanoparticle combinations to come to conclude that the movement of nanoparticles is indeed moved by optical scattering force.

However, there are few weak points of this manuscript.

First, the calculation of optical force and maximum particle velocity with modified Stoke's Law is oversimplified and may not be necessary. The particle is very close to the surface and the surfactant property changes under light illumination. The simple model used by the authors will not be able to capture these many parameters involved. This may be why there is 2~3 orders of magnitude difference between the measurement results and their theoretical model.

Second, based on the Stokes' Law, any accelerated particle movement will reach to a terminal velocity in less than 1 usec for submicron sized particles. Therefore, there should be no acceleration term in the equation $m\ddot{x} = F_{res} + F_{opt}$ when you measure the system at a time scale longer than a usec after your force applied. It will be misleading to use particle "acceleration" to describe the phenomena measured. The particle velocity measured in msec time scale using fast CCD in this paper should be the result of the balance between the optical force and the transient fluid or surfactant properties changed by heat.

Third, the averaged particle position deviation is ~200 nm. How can this resolution allows the control of assembled nanoscale particles. For example, can this technique be reliably used to control the 15nm gap spacing of a dimer of AuNP?

Reviewer #1:

This paper presents a novel mechanism for submicron particle manipulation. It combines a phase change of a material by optothermal heating with optical scattering forces. This interesting mechanism should be of interest to researchers in this field.

Overall, this is an interesting, well-written, and comprehensive paper. However, some further revisions can strengthen this paper further:

Answer: Thank you very much for the reviewer's comments. We have made point-to-point responses to the insightful comments:

1. Since the mechanism pushes a particle under manipulation by optical scattering force, the manipulation is inherently unstable. Please include a better characterization of the accuracy of the particle movement. One possibility would be to make a measurement similar to "optical trapping stiffness" that is regularly characterized for optical tweezers. Another suggestion is to determine the difference between the laser movement vector and the particle movement vector at various points along a trajectory, which could be done by analysing the existing recordings of particle movement.

Answer: Thank you for the reviewer's comments. We agree that a better characterization of the accuracy of particle movement is needed, and we appreciate the reviewer's suggestions. In the revised manuscript, we carefully analysed the recorded videos to determine the difference between the laser movement and the particle movement vectors along the trajectory. During the manipulation process, the trajectories of the particle and the laser beam almost overlap (Fig. R1, also see Supplementary Fig. 8), demonstrating that the particles can be efficiently manipulated along the laser direction. The schematic of the comparison of these two vectors is shown in Fig. R2 (also see Figure 2d in the revised

manuscript). We define the manipulation efficiency Q by the dot product of unit vectors along the laser movement vector E_l and the particle movement vector E_p . The calculated Q ranges from ~ 0.6 - 0.8 for all colloidal particles, which shows efficient manipulation. We have discussed these analyses in detail in the revised manuscript:

Figure R1. Trajectories of the particles and laser beam recorded in Supplementary Movie 1. **a**, 200 nm AuNP; **b**, 80 nm AuNP, moving forth; **c**, 80 nm AuNP, moving back; **d**, 100 nm AgNP; **e**, 500 nm SiNP. The manipulation efficiencies of each particle are also presented in the figures.

Figure R2. Comparison of the laser movement vector E_l and the particle movement vector E_p at two successive frames ($t = t_1, t_2$). θ is denoted as the angle between E_l and E_p .

Page 9, Lines 160-169: “We further characterize the manipulation efficiency by analysing the video recordings of the particle movement. During the manipulation process, the trajectories of the particles and the laser beam almost overlap (Fig. R1, also see Supplementary Fig. 8), which shows that the particles can be efficiently manipulated along the laser direction. To quantify the manipulation efficiency, we examined the difference between the laser movement vector E_1 and the particle movement vector E_p , as sketched in Fig. R2 (also see Figure 2d in the revised manuscript). The accuracy of the particle movement can be characterized by the dot product of unit vectors along E_1 and E_p . We define a manipulation efficiency Q as the average $\cos\theta$ over a full manipulation trajectory:

$$Q = \langle \cos\theta \rangle = \left\langle \frac{E_1 \cdot E_p}{|E_1||E_p|} \right\rangle \quad (3)$$

where θ is the angle between E_1 and E_p . The calculated Q ranges from ~0.6-0.8 for the recorded videos, indicating high-efficient manipulation of all kinds of colloidal particles.” was added.

2. Equation 1 in the paper is Stokes' Law, but this form of the equation assumes an unbounded fluid. This equation needs a correction factor when the particles are close to a surface, as is the case in this paper. In addition, it is not clear how this equation is applied to find the resistance force. Is it used only for the portion of the particle cross-section traveling in the CTAC layer?

Answer: Thank you for the reviewer’s comments. During the OPN manipulation, the CTAC surrounding the nanoparticles undergoes a localized phase transition to a “quasi-liquid” structure, and the nanoparticles are partially immersed into this quasi-liquid thin film. The resistant forces for a moving particle which is partially immersed in the liquid film can be calculated by a modified Stokes equation¹⁻³:

$$F_{res} = 6\pi\eta R f_d \mathbf{v} \quad (\text{R2, also see Eq 1 in the revised manuscript})$$

where η is the viscosity of CTAC in its “quasi-liquid” phase, R is the particle radius, v is the velocity of the particle. Compared to normal Stokes’ Law, the drag coefficient f_d , an additional factor, is accounted for, representing the effect of the air-liquid interface. In the revised manuscript, we thoroughly discussed the calculation of the resistant forces. First, we took the tilted SEM images to confirm the partial immersion of the particles into the CTAC film. Then, we performed molecular dynamics (MD) simulations to calculate the viscosity of CTAC at 450 K. Finally, the drag coefficient was determined based on the reported results². We discussed the detailed calculation of resistant forces of nanoparticles in Supplementary Note 3:

Page 8, Lines 147-148: “The resistant force can be evaluated according to” ***was replaced by*** “In our case, the particles are partially immersed into the CTAC film (Fig. R3, also see Supplementary Fig. 7), for which the resistant force can be evaluated according to”

Page 8, Line 152: “The detailed calculation of the resistant force can be found in Supplementary Note 3.” ***was added.***

Supplementary Information, Pages 13-16, Lines 145-201:

“Supplementary Note 3. Calculating resistant forces in OPN

During OPN manipulation, colloidal particles are partially immersed into the quasi-liquid CTAC film (Fig. R3, also see Supplementary Fig. 7). Thus, the resistance force to the colloidal particle during the manipulation can be modelled according to Equation R2. To calculate F_{res} , we determined η and f_d as discussed below.

Fig. R3. 45° titled SEM images of 200 nm AuNPs. a, AuNPs on a CTAC film; **b**, AuNPs on a glass substrate. There is an obvious immersion line for the AuNPs on a CTAC film (red dashed lines in **a**), while for the glass substrate, the AuNPs maintains its spherical shape. These results clearly show that the particles are partially immersed into CTAC film during OPN manipulation.

3.1 MD simulation of the viscosity of the quasi-liquid CTAC

The viscosity of CTAC in its quasi-liquid state was calculated by molecular dynamics (MD) simulations with the LAMMPS package⁴. The velocity Verlet algorithm is employed in integrating equations of motion, and the time step is 0.25 fs. Initially, the isothermal–isobaric (NPT) ensemble is employed to reach the required temperature and pressure (1 atm). Then, the system is equilibrated under the canonical ensemble (NVT) with the Langevin heat reservoir at the target temperature for 0.5 ps, followed by relaxation under a microcanonical ensemble (NVE) for 0.2 ns. Finally, a production step of 10 ns was adopted under the NVE condition, during which the pressure tensor was calculated every 10 fs to obtain PACF. For each case, four independent simulations were performed with different initial atom velocity assignments, implemented by using different seeds for random number generation. Averaged values

were obtained to improve the reliability of the simulation results. The SHAKE algorithm was employed to fix geometries of the water molecules and partial bonds of CTAC molecule⁵. Long-range electrostatic interactions were counted using the particle-particle particlemesh method⁶ with a precision of 10^{-6} .

Figure R4. MD simulation of CTAC. **a,b**, Snapshots of CTAC in MD simulations at **(a)** 300 K and **(b)** 450 K. **c**, Calculated viscosity and normalized pressure tensor autocorrection function (PACF) of CTAC at 450 K.

During the OPN manipulation, the temperature of CTAC surrounding the AuNP exceeds 450 K (Fig. 1d). Thus, we simulated the structure and viscosity of CTAC at 450 K. An order-disorder phase transition of CTAC can clearly be observed when temperature increases from 298 K to 450 K (Fig. R4a,b, also see Supplementary Fig. 12), confirming the formation of quasi-liquid CTAC. The result is consistent with the reported publications¹⁵. The viscosity of the quasi-liquid CTAC is calculated based on Green-Kubo linear response theory by the integral of pressure tensor autocorrelation function (PACF) via¹⁶

$$\eta = \frac{V}{k_B T} \int_0^\tau \langle P_{\alpha\beta}(0) P_{\alpha\beta}(t) \rangle dt \quad (\text{R3})$$

where V is the system volume, k_B is the Boltzmann constant, T is the temperature and τ is correlation time. The angle bracket denotes time correlation function, also interpreted as ensemble averaging. $P_{\alpha\beta}$ is

an off-diagonal ($\alpha, \beta = x, y, z. \alpha \neq \beta$) element of the pressure tensor, which for an N -particle system is calculated by

$$P_{\alpha\beta} = \frac{1}{V} \sum_{i=1}^N (m_i v_{i\alpha} v_{i\beta} + r_{i\alpha} f_{i\beta}) \quad (\text{R4})$$

where m_i , v_i , r_i and f_i are the mass, velocity, position, and force of the atom i , respectively. The simulated viscosity at 450 K was calculated to be 8.7 mPa·s (Fig. R4c, also see Supplementary Fig. 12), which was used for calculation of the resistant forces.

3.2 Determination of the drag coefficient

According to the theory developed by Danov *et al*¹¹, when a colloidal particle is half-immersed in a thin liquid film, the drag coefficient f_d is only a function of the contact angle α (Fig. R5, also see Supplementary Fig. 13) and the surface viscosity. In our system, CTAC has a free surface (no liquid-liquid interface), therefore, the surface viscosity = 0. In this case, f_d is almost independent of the contact angle¹¹. When $\alpha = 90^\circ$, f_d equals 0.5 corresponding to the half-immersed case. In our case, the contact angle is larger than 90° , and the corresponding f_d is smaller than 0.5. Based on the results by Danov *et al*¹¹, the drag coefficient in our case was estimated to be ~ 0.4 for the calculation of resistant forces. As an example, we calculated the resistant force for 200 nm AuNPs based on the experimental velocities. At the optical power of 1.4 mW, the measured velocity is $\sim 70 \mu\text{m/s}$, and the calculated resistant force is $\sim 0.35 \text{ pN}$, which is in similar range of the optical forces.

Figure R5. Geometry of our system to define the contact angle α ” was added.

3. In Figure 2b, please change the x-axis to a narrower range, so that the change in velocity when the laser is switched on is more apparent. (There is a steep slope to the velocity, but it should be finite.)

Answer: Thank you for the reviewer’s comments. We have changed the x-axis to a narrower range in the revised manuscript as suggested (Fig. R7, also see Fig. 2b). We have changed the x-axis to a narrower range for Supplementary Fig. 6 as well.

Figure R7. The measured X position, Y position, and speed v of a 300 nm AuNP under the laser irradiation of 1.40 mW as a function of time t . The solid green arrow at $t = \sim 2.7$ s indicates the instant when the laser is turned on.

Reviewer #2:

This paper describes a method terms, termed optothermally-gated photon nudging (OPN), for inducing motion of colloidal particles and nanowires on a solid substrate. This solid-phase optical micromanipulation exploits placement of a thin surfactant layer between the particles to be organised and a substrate. This layer acts as an optothermal gate to optically modulate the particle-substrate

interaction. The authors claim this breaks "the limitations of conventional optical tweezers to liquid, vapor or vacuum environments".

The particular modulation of interfacial interactions as proposed to lower the friction forces via the thin surfactant layer is introduced between the particles and the glass substrate, The optical heating of the particles, the friction of the particle and surfactant is dramatically reduced due to the phase transition of the surfactant layer, which allows the manipulation of particles with optical scattering forces.

The paper has some interest but falls well short of showing the compelling data or advance over previous literature that would warrant publication in Nature Communications. Some points I would raise is support of my conclusions as well as more general points regarding the paper are below:

Answer: Thank you so much for the reviewer's comments. In this manuscript, we report OPN as a novel type of optical manipulation techniques, which is also the first time to demonstrate a solid-phase optical manipulation technique. We designed a new platform to modulate the particle-substrate interfaces to achieve on-demand manipulation and patterning of a wide range of colloidal particles on solid substrates by coordinating optothermal effects and optical scattering forces. The new mechanisms demonstrated here is interesting for the future development in this field. This new technique will provide the potential to overcome the limitations in current optical manipulation techniques, which are commonly operated in solution environments. By developing a solid-phase technique, we can avoid the undesired pattern collapse in liquid, overcome the Brownian motion of nanoparticles, and provide a reconfigurable patterning technique for particle assembly and organisation. We believe this method will offer a new solution for dynamic assembly of colloidal particles on a solid substrate, the exploration of coupling between colloidal particles, and the fabrication of active colloidal functional devices. Therefore,

we believe this work is appropriate to be published in Nature Communications. We have made point-to-point responses to the reviewer's comments:

1) This paper seems to me a variant of their recent Nature Photonics (<https://www.nature.com/articles/s41566-018-0134-3>, not cited!) where a cationic surfactant, cetyltrimethylammonium chloride (CTAC) is added to the nanoparticle colloid. why is this work not properly described nor placed in context?

Answer: Thank you for the reviewer's comments. Although CTAC is used in both this work and our Nature Photonics paper⁷, these two works are completely different in every aspect:

1. In this work, we use CTAC as a thin solid layer and exploit its localized order-disorder phase transition to open the optothermal gate; while in the Nature Photonics paper, CTAC is added into the solution, in which the spatial separation of CTA⁺ micelles and Cl⁻ ions are exploited to create a thermoelectric field. Since our technique is based on the phase transition properties of the materials, we can also achieve OPN on other surfactants (e.g., SDS) and other materials with similar physical properties, such as PMMA.
2. In this work, we rely⁷ on optical scattering force as the driving force to push and manipulate particles, while in the Nature Photonics paper, thermoelectric forces are responsible for the trapping of particles.
3. In this work, we exploit optothermal effects of colloidal particles (or external optothermally responsive materials) to induce the phase transition of CTAC to modulate the particle-substrate interactions, while in the Nature Photonics paper, we exploit optical heating of an optothermal substrate to generate a temperature gradient in solution to induce the thermophoretic migration of colloidal species.

4. In this work, particles are manipulated on a solid substrate, while in the Nature Photonics paper, particles are manipulated in solution.
5. In this work, we developed in situ optical spectroscopy to characterize the optical properties of colloidal particles without the need of keeping the laser on, while in the Nature Photonics paper, laser is necessary to keep the particle at a specific position for the measurement of scattering spectra.
- We summarized these differences in the Table R1. Based on these differences, we believe this work is a novel technique rather than a variant of our recent Nature Photonics paper.

Table R1. Summary of the comparisons of this work and our recent Nature Photonics paper.

	This work	Nature Photonics paper⁷
Form of CTAC	Solid layer	Added to solution
Function of CTAC	Optothermal gate to modulate particle-substrate interactions	To generate a thermoelectric field
Driving forces	Optical scattering forces	Thermoelectric forces
Role of optical heating	Induce the phase transition of CTAC	Generate a temperature gradient to induce the thermophoretic migration
Working environment	On solid substrates	In solution
Is laser necessary to maintain particle locations	No	Yes

2) what is the advantage over

- large area evanescent wave large area traps with added single beam tweezers from above? - e.g adding tweezers to a system like Phys. Rev. B 73, 085417 (2006) or

-light induced dielectrophoretic traps (Ming Wu group)

- other optothermal methods, eg. Chen, Hongtao & Gratton, Enrico & Digman, Michelle. (2016). Self-assisted optothermal trapping of gold nanorods under two-photon excitation. see Methods and Applications in Fluorescence. 4. 10.1088/2050-6120/4/3/035003.

Answer: Thank you for the reviewer's comments and suggested references. We have checked the suggested publications⁸⁻¹⁰ and related works^{11,12} carefully. The advantages and novelties of this work are listed as below:

1. We achieved the nanomanipulation on a solid substrate in air not in solution. It is impossible to achieve manipulation on a solid substrate in air using SPP as mentioned in Phys. Rev. B 73, 085417 (2006) or using dielectrophoretic trapping as mentioned in Ming Wu's group due to the strong van der Waals force between the particle and the substrate¹³.
2. This work features simpler optics compared to the references above. For example, ref. 8 and ref. 12 require the optical setup to excite the surface plasmon polariton to enhance the trapping and assembly; ref. 9 needs an external A.C. electrical field to induce the particle manipulation; and ref. 10 requires the optical setup for two-photon excitation. In this work, a simple optical setup (a single laser beam) is used without any rigorous requirement on optical alignment.
3. In this work, we can achieve the precise manipulation of colloidal particles at a single-particle resolution with nanoscale accuracy, while in the references mentioned above, the manipulation of a single particle remained to be a challenge.
4. In this work, the patterned structures will maintain their position after manipulation. In the references, the manipulation experiments are operated in solution, and the assembled structure cannot be maintained after the laser is turned off.

Despite the advantages mentioned above, we also realized that this work still has limitations on the throughput and manipulation efficiency, which could be further improved by future work. We discussed

the limitations in detail and proposed possible solutions in the Discussion section in the revised manuscript:

Page 18, Lines 318-325: “Future efforts can be made to further enhance the strengths of the solid-phase optical manipulation technique. One can optimize the optics to achieve a more efficient operation. For instance, oblique incidence of the laser can take advantage of photon momentum along the direction of beam propagation, which could enhance both amplitude and directional control of the driving forces. While OPN offers the opportunity to manipulate colloids at single-particle resolution, it suffers from relatively low patterning throughput, which is primarily limited by its serial and manual control. The implementation of a light spatial modulator with a digital feedback control will open up the possibilities for automatic and parallel manipulation to significantly boost the production output.” **was added.**

3) equation (1), line 154, can't be correct...surely we need to use the Faxen correction given the close proximity of the particles to the surface? What in fact is this distance and how is they determined?

Answer: Thank you for the reviewer's comments. We have confirmed that particles are partially immersed into CTAC quasi-liquid film and applied the modified Stokes' law¹⁻³ to calculate the resistant force. The Faxen correction factor was determined according to the developed theory in Ref.2. The detailed discussion on the calculation of the resistant forces can be found in the previous pages in this letter (Pages 3-7, also see Supplementary Note 3).

4) A position accuracy of ~80 nm was achieved, as indicated by the shaded area seems poor for the SiNPs with a diameter of 500 nm placed into a straight line...~15% accuracy? This will have to be improved for any reproducible particle organisation

Answer: Thank you for the reviewer's comments. The position accuracy in this work was mainly limited by the optical imaging due to diffraction limit. In principle, our technique can manipulate colloidal particles into any desired positions with much higher accuracy. This limitation can be overcome with the assistance of advanced instruments to improve the imaging. In addition, during our experiments, we only relied on our naked eyes to estimate the right positions of the nanoparticles. Thus, the patterning accuracy can also be further improved with predesigned markers or software to precisely determine the target positions. We have demonstrated that the position accuracy can be improved to ~20-30 nm with the assistance of software to define the target lines for particles. We have discussed this problem in the revised manuscript and Supplementary Note 4 in more details:

Page 11, Lines 199-205: “It should be noted that during the experiments, we only relied on our naked eyes to estimate the right positions of the nanoparticles. Thus, we envision that the patterning accuracy can be further improved with the assistance of software or predesigned markers to determine the target positions.” **was replaced by** “It should be noted that the current position accuracy is primarily limited by the diffraction barrier in optical microscopy. Additionally, during the manipulation experiments, we only relied on our naked eyes to estimate the positions of the nanoparticles in the optical images. Thus, we envision that the particle-patterning accuracy can be further improved with advanced imaging, analysis and tracking of particles with higher precision. For instance, we have achieved an improved position accuracy of ~20 nm by using the imaging software to define a target line along which the particles will be aligned (Supplementary Note 4).”

Supplementary Information, Pages 16-17, Lines 203-213:

“Supplementary Note 4. Improving patterning accuracy with the assistance of the software

The patterning accuracy of OPN is primarily limited by the optical diffraction limit. Additionally, we only relied on our naked eyes to estimate the positions of the nanoparticles during the experiments. Here, we demonstrate that it is possible to apply the predesigned markers or imaging software to define the target lines for particle manipulation. As shown in Supplementary Fig. R8a (also see Supplementary 14), we used the imaging software to define the target line and moved five SiNPs to the lines. With the assistance of the target lines, the average position error was reduced to less than ~ 30 nm (Fig. R8b,c, also see Supplementary 14).

Figure R8. Precise patterning of 500 nm SiNPs with the the assistance of the software . a,b, Dark-field images (a) and SEM images (b) of two patterned lines of 5 SiNPs. c, The corresponding position errors of each particles in (b). All Scale bars are $5 \mu\text{m}$.” was added.

5) fig 3 is really for such a small number of particles. it would need 2-3 orders of magnitude more to be convincing for real self organisation

Answer: Thank you for the reviewer’s comments. Yes, we agree for self-organisation more than hundreds of particles are needed, such as self-assembly¹⁴. In this work, what we demonstrated is particle manipulation on solid substrates, not self-organisation of particles. We have achieved controlled

manipulation and patterning of 7 and 9 particles into a line and a 3x3 array, respectively, to demonstrate the patterning accuracy of this technique. This number is surely not convincing for self-organisation; but we believe it is sufficient for light-controlled manipulation. More complex structures and patterns can be generated at single-particle levels.

6) The CTAC layer is removable without altering the particle positions after OPN patterning, as CTAC might be best to avoid for some applications. This should be in the main text and discussed in more detail

Answer: Thank you for the reviewer's suggestion. We have moved the removal of CTAC to the main text and discussed it in detail in the revised manuscript:

Page 11, Lines 205-206: “Additionally, the CTAC layer is removable without altering the particle positions after OPN patterning (see Supplementary Note 4), in case CTAC is undesirable for some practical applications.” **was replaced by** “It is also worth noting that the CTAC layer is removable without altering the particle positions after OPN patterning, which will be discussed in detail later.”

Page 15, Lines 264-272: “We further showed that the CTAC layer can be readily removed without destroying the existing particle patterns by simply soaking the sample in water or isopropyl alcohol (IPA) for ~2 minutes. As shown in Figs. 5a and b, the positions of the colloidal particles remained the same after the removal of CTAC. Meanwhile, the measured scattering peak of 100 nm AuNPs showed an obvious blueshift from ~580 nm to ~545 nm (Fig. 5c). This blueshift revealed a refractive-index change in the particle surrounding, which confirmed the successful removal of CTAC (the refractive index of CTAC is 1.38). The ability to remove CTAC after OPN manipulation can avoid any undesirable effects

of CTAC in some of applications of the patterned particles, including chemical and biological sensing.”
was added.

Figure R9. Removal of CTAC without altering the particle positions. **a,b,** Optical images of a pentagon pattern composed of five 100 nm AuNPs (**a**) before and (**b**) after the removal of CTAC. Two same white dashed pentagons are added to help distinguish the positions of AuNPs. **c,** The scattering spectra of 100 nm AuNPs measured before and after the removal of CTAC. The shaded area indicates the standard deviation of the peak position. The inset shows the corresponding dark-field images. Scale bars: **a,b,** 10 μm ; inset in **c,** 2 μm .

7) Reconfigurable patterning of four 300 nm AuNPs is interesting but hardly enough for proving organisation

Answer: Thank you for the reviewer’s comments. As we stated before, our technique is light-directed manipulation rather than self-organisation of particles. While we agree reconfigurable patterning of four 300 nm AuNPs is hardly enough for proving dynamic self-organisation, we believe it is enough to demonstrate the active manipulation of particles on solid substrates.

8) what is the optimum choice of laser wavelength for optothermal gate?

Answer: Thank you for the reviewer’s insightful comments. To open the optothermal gate, we need the laser to heat the particle or the external optothermal substrate to induce the phase transition of CTAC. Therefore, to efficiently open the optothermal gate, the optimal working wavelengths for OPN should be selected to match the absorption of the colloidal particles or the external optothermal substrate. We have further performed control experiments to demonstrate the importance to choose a laser wavelength which matches with the absorption of particles. The detailed discussion can be found in the Supplementary Note 7:

Supplementary Information, Pages 19-20, Lines 251-269:

“Supplementary Note 7. Optimal wavelength for OPN

The working wavelengths for OPN can be properly optimized to reduce the optical power needed to open the optothermal gate and improve the performance. Take the manipulation of 80 nm AuNPs as an example, in which both 532 nm and 660 nm lasers were tested for OPN experiment. The results showed that the 80 nm AuNPs can be readily manipulated by 532 nm laser with a low power of 1.0 mW. In contrast, 660 nm laser doesn’t work even with a much higher power of 2.5 mW. This is because 80 nm AuNPs have much stronger absorption at 532 nm than that at 660 nm (Fig. R10c, also see Supplementary Fig. 17). The simulated temperature also shows that the temperature surrounding the particle reaches more than 450 K for the 532 nm laser, opening the optothermal gate, while a 660 nm laser produces a temperature of only ~330 K (Fig. R10a,b, also see Supplementary Fig. 17). Therefore, the optimal working wavelengths for OPN can be selected to match the absorption cross-section of the colloidal particles. In addition, as we demonstrated, it is also possible to introduce other optothermal materials (such as gold nanoislands) as external heating sources to induce the phase transition of CTAC,

thus, the optimal wavelength, in this case, should be chosen based on the absorption of the external optothermal materials to open the optothermal gate.

Figure R10. Optimal working wavelength for OPN. a,b, Simulated temperature distribution around an 80 nm AuNP. (a) 532 nm laser with an incident power of 1 mW; (b) 660 nm laser with an incident power of 2.5 mW. Scale bars: 80 nm. **c,** Simulated absorption cross-section for 80 nm AuNPs. The green and red dashed lines stand for the wavelengths of two lasers used.” *was added.*

9) fabrication of active devices and functional components is mentioned. The authors need to explain in some detail what they might be

Answer: Thank you for the reviewer’s comments. Since most functional colloidal devices are operated on solid substrates, one of the unique advantages of this work is to provide a method for reconfigurable assembly and patterning of colloidal particles into different configurations directly on a solid substrate. In combination with the non-invasive operation, our OPN is well-suited for the fabrication of colloidal devices with tunable responses and functions. For instance, the dynamic assembly of colloidal particles can be functioned as reconfigurable optical nanocircuits¹⁵ or reconfigurable chiral metamolecules and related chiroptical devices^{16,17}. In addition, the manipulation of gold nanowires can also be exploited to develop active plasmonic waveguide to control the steering direction of light^{18,19}. We added more details in the revised manuscript as suggested:

Page 15, Lines 262-264: “This non-invasive operation is highly desired and advantageous in the fabrication of active devices and functional components.” ***was replaced by*** “This non-invasive operation is highly desired and advantageous in the fabrication of functional components and devices such as reconfigurable optical nanocircuits and active plasmonic waveguides.”

10) as inherently still relying on optical scattering which is still pN of force, surely this approach is too slow/weak to be of use for larger scale organisation?

Answer: Thank you for the reviewer’s comments. In this work, we rely on optical scattering forces to manipulate colloidal particles. The magnitude of scattering force is in pN, which is of similar magnitude to the gradient forces in optical tweezers while our operation power is much less than that of optical tweezers. Also, since we are primarily demonstrating the manipulation of colloidal nanoparticles with the diameter less than 500 nm. The driving force will scale up with the size of the colloidal particles. In addition, we have demonstrated that scattering force can be exploited for efficient manipulation as shown in the previous pages in this letter (Pages 1-3). Meanwhile, we also discussed in detail the current limitations and potential solutions to achieve larger scale patterning of colloidal particles in the Discussion section, as shown in Page 13.

Overall, this is really a modest improvement and the data show are not of the level for a Nature Comm. A revised paper would be good for an ACS journal or App Phys Lett

Answer: Thank you for the reviewer’s comments. In this work, we report for the first time the solid-phase optical manipulators, which can achieve non-invasive and contactless particle manipulation on solid substrates with a single laser beam. We also present novel mechanisms to simultaneously harness both optothermal effects and optical scattering forces – which are considered as losses in conventional

optical tweezers – for particle manipulation. Furthermore, we show new possibilities to dynamically assemble and pattern colloidal particles directly on substrate, without the need to compete with Brownian motion. Taking advantage of the in situ optical microscopy, this technique can be applied for the investigation of coupling between colloidal particles at the nanoscale. In summary, this work presents a novel solid-phase optical manipulation technique which will provide new possibilities and advances from the current techniques. Thus, we believe this work is appropriate for Nature Communications.

Reviewer #3:

This manuscript presents an innovative concept that allows a light beam to control and pattern nanoscale particles. Their impact comes from the idea of using a layer of surfactant molecules that can change between solid and fluid phases at locations under light illumination and heat generation. Without light illumination, the nanoparticles are stably anchored on the surface. Under illumination, nanoparticles are pushed to move on a 2D surface due to a combination effect of optical scattering forces and molten surfactant.

This is the most unique part of this work compared to other optical manipulation mechanisms that need to always compete with Brownian motion and have difficulty to maintain particle locations once light beams are removed. To understand the true working mechanism behind this interesting phenomenon, the authors have done a nice comparison of different kind of substrate and nanoparticle combinations to come to conclude that the movement of nanoparticles is indeed moved by optical scattering force.

However, there are few weak points of this manuscript.

Answer: Thank you so much for the reviewer's careful reading and positive comments. We have made point-to-point responses to the weak points raised by the reviewer:

First, the calculation of optical force and maximum particle velocity with modified Stoke's Law is oversimplified and may not be necessary. The particle is very close to the surface and the surfactant property changes under light illumination. The simple model used by the authors will not be able to capture these many parameters involved. This may be why there is 2~3 orders of magnitude difference between the measurement results and their theoretical model.

Answer: Thank you for the reviewer's comments. We agree that the calculation of theoretical maximum speed is not necessary. The theoretical maximum speed was simply calculated by only considering optical force and resistant force, with an assumption that the laser will always be at the position where the optical force is at its maximum. However, this assumption is too ideal to be realized. During the real manipulation, the laser beam is moved to push the colloidal particles. Currently, due to the difficulty in precisely controlling the relative position of laser beam and target particle, the average speed of particle manipulation is limited. However, we propose that with integration of digital controls digital operation and feedback controls, the manipulation speed can be further improved. We have deleted the original Supplementary Note on the calculation of maximum speed and revised the manuscript accordingly:

Pages 5-6, Lines 95-97: "It should be noted that the speed of the OPN manipulation was limited by the manual operation in this scenario. However, with automatic digital controls, the theoretical maximum speed can reach up to ~100 $\mu\text{m/s}$, which promises great potential for efficient manipulation (see Supplementary Note 1)." **was replaced by** "It should be noted that the speed of the OPN manipulation

was limited by the manual operation in this scenario. With automatic digital operation and feedback controls, the manipulation speed could be further improved.”

Second, based on the Stokes’ Law, any accelerated particle movement will reach to a terminal velocity in less than 1 usec for submicron sized particles. Therefore, there should be no acceleration term in the equation $m\ddot{x} = F_{res} + F_{opt}$ when you measure the system at a time scale longer than a usec after your force applied. It will be misleading to use particle “acceleration” to describe the phenomena measured. The particle velocity measured in msec time scale using fast CCD in this paper should be the result of the balance between the optical force and the transient fluid or surfactant properties changed by heat.

Answer: Thank you for the reviewer’s thoughtful comments. We agree that the acceleration period is too short and the measurement of particle “acceleration” in msec time scale is misleading. As the reviewer suggested, the particle velocity measured using fast CCD in this work should be the result of the balance between the optical force and the resistant forces by the surfactant. Therefore, in the revised manuscript, we replaced the term “acceleration” with “measured maximum speed”. We also applied the physical model to simulate the maximum speed with the same timestep, and the results match very well with the experiments (Figure R11, also see Fig. 2c). We have discussed this in detail in the revised manuscript:

Page 7, Line 128: “acceleration” ***was replaced by*** “velocities”

Pages 7-8, Lines 130-131: “The measured particle velocity is the result of the balance between the optical driving force and the resistant force by surfactant.” ***was added.***

Page 8, Line 138: “acceleration” ***was replaced by*** “maximum velocities”

Page 9, Line 157: “acceleration” ***was replaced by*** “velocity”

Figure R11. Measured maximum speed of 200 nm and 300 nm AuNPs as a function of incident power. The solid lines show the corresponding modelled data.

Third, the averaged particle position deviation is ~ 200 nm. How can this resolution allows the control of assembled nanoscale particles. For example, can this technique be reliably used to control the 15nm gap spacing of a dimer of AuNP?

Answer: Thank you for the reviewer's comments. As stated before, the particle position accuracy is mainly limited by the optical imaging due to the diffraction limit. In this case, it is not possible to precisely control the assembled nanoscale particles by distinguishing the dark-field images. However, taking advantage of the in situ optical spectroscopy, we can deduce the interparticle gap by measuring the near field coupling between colloidal particles. As the reviewer suggested, we demonstrated the controlled fabrication of a 100 nm Au dimer with a gap of ~ 15 nm. We fabricated two Au dimers with the same two 100 nm AuNPs. While we cannot distinguish the gap based on the optical images, we can measure the distinct scattering spectra of these two dimers. One dimer shows single peak without any near field coupling, and the other dimer shows the peak splitting behaviour corresponding to a dimer with a gap of ~ 15 nm. The SEM image further confirmed the reliable fabrication of the Au dimer with a ~ 15 nm gap. The detailed discussion is presented in Supplementary Note 6:

Page 16, Lines 285-287: “It is worth noting that although the sub-wavelength interparticle gap cannot be distinguished in optical images due to the diffraction limit, OPN can reliably fabrication Au dimers with any desired gap by taking advantage of the *in situ* optical spectroscopy (Supplementary Note 6).”
was added.

Supplementary Information Pages 18-19, Lines 229-249:

“Supplementary Note 6. Reliable fabrication of Au dimers with ~15 nm gap

The diffraction limit in optical microscopic imaging has prevented the measurement of the distance between two close nanoparticles. In this experiment, two 100 nm AuNPs (Fig. R12d, also see Supplementary Fig. 16) were assembled into a Au dimer. Fig. R12a,b (also see Supplementary Fig. 16) show the dark-field optical images of the Au dimer composed of two same 100 nm AuNPs with different interparticle distances. It is challenging to distinguish the gap between these two AuNPs based on the optical images. However, taking advantage of our *in situ* optical spectroscopy, one can easily obtain the scattering spectra of the Au dimers. As shown in Fig. R12e (also see Supplementary Fig. 16), the scattering spectrum of the dimer in Supplementary Fig. R12a (also see Supplementary Fig. 16) shows a single peak at ~585 nm, which is consistent with that of single AuNPs. The spectrum can also be well-fitted by a single-peak Lorentz function. The result reveals that there is no near-field coupling between two AuNPs, which indicates two AuNPs are separated by a large distance. In contrast, the dimer in Fig. R12b (also see Supplementary Fig. 16) shows two split peaks at ~575 nm and ~635 nm (Fig. R12e, also see Supplementary Fig. 16), which demonstrate the near-field coupling behavior between these two AuNPs. The peak splitting results are consistent with those shown in Fig. 5. The SEM image further confirmed a Au dimer with a gap of ~ 15 nm (Fig. R12c, also see Supplementary Fig.

16). In summary, we can reliably fabricate the Au dimer with the desired gap by analyzing the scattering spectra with the help of *in situ* optical spectroscopy.

Figure R12. Reliable fabrication of Au dimer with a gap of ~15 nm. a,b, Optical images of the Au dimer composed of two same 100 nm AuNPs with different interparticle distances. **c,** The SEM images of the Au dimer in **(b)**. **d,** The scattering spectra of the two single AuNPs. **e,** Scattering spectra and the fitting of the Au dimers in **(a)** and **(b)**.” was added.

Additional revisions:

We also made additional revisions to improve the fluency of the manuscript, as listed below. The additional changes are also highlighted in the revised copy of manuscript.

Page 2, Line 18: “,” was added.

Page 2, Line 19: “Briefly” was replaced by “In summary”.

Page 2, Lines 24-25: “With” was replaced by “Along with”.

Page 3, Line 43: “reconfigurable patterning is impossible in this case” *was replaced by* “reconfigurable patterning becomes impossible”.

Page 4, Line 55: “Briefly” *was replaced by* “In short”.

Page 4, Line 55: “which allows” *was replaced by* “allowing”.

Page 4, Line 63: “for the” *was replaced by* “with”.

Page 4, Line 70: “For the purpose of our demonstration” *was replaced by* “For demonstration”.

Page 4, Line 73: “allow” *was replaced by* “allows”.

Page 7, Line 113: “Force analysis” *was replaced by* “Characterizations of OPN manipulation”.

Page 8, Line 134: “the” *was added*.

Page 8, Line 136: “speed” *was replaced by* “speeds”.

Page 8, Line 144: “Fig. 2e” *was replaced by* “Fig. 2f”.

Page 9, Line 158: “Fig. 2f” *was replaced by* “Fig. 2c”.

Page 13, Line 226: “maneuvering spherical” *was added*.

References

- 1 Kralchevsky, P. A. & Nagayama, K. Capillary interactions between particles bound to interfaces, liquid films and biomembranes. *Adv. Colloid. Interfaces* **85**, 145-192 (2000).
- 2 Danov, K., Aust, R., Durst, F. & Lange, U. Influence of the surface viscosity on the hydrodynamic resistance and surface diffusivity of a large brownian particle. *J. Colloid. Interface Sci.* **175**, 36-45 (1995).
- 3 Petkov, J. T. et al. Precise method for measuring the shear surface viscosity of surfactant monolayers. *Langmuir* **12**, 2650-2653 (1996).
- 4 Plimpton, S. Fast parallel algorithms for short-range molecular dynamics. *Journal of Computational Physics* **117**, 1-19 (1995).
- 5 Hockney, R. W. & Eastwood, J. W. *Computer simulation using particles*. (crc Press, 1988).
- 6 Sun, H. Compass: An ab initio force-field optimized for condensed-phase applications overview with details on alkane and benzene compounds. *J. Phys. Chem. B* **102**, 7338-7364 (1998).
- 7 Lin, L. et al. Opto-thermoelectric nanotweezers. *Nat. Photonics* **12**, 195-201 (2018).
- 8 Garcés-Chávez, V. et al. Extended organization of colloidal microparticles by surface plasmon polariton excitation. *Phys. Rev. B* **73**, 085417 (2006).
- 9 Chiou, P. Y., Ohta, A. T. & Wu, M. C. Massively parallel manipulation of single cells and microparticles using optical images. *Nature* **436**, 370-372 (2005).
- 10 Chen, H., Gratton, E. & Digman, M. A. Self-assisted optothermal trapping of gold nanorods under two-photon excitation. *Methods and Applications in Fluorescence* **4**, 035003 (2016).
- 11 Kang, Z. et al. Trapping and assembling of particles and live cells on large-scale random gold nano-island substrates. *Sci. Rep.* **5**, 9978 (2015).
- 12 Brasiliense, V. et al. Light driven design of dynamical thermosensitive plasmonic superstructures: A bottom-up approach using silver supercrystals. *ACS Nano* **12**, 10833-10842 (2018).
- 13 Urban, A. S., Lutich, A. A., Stefani, F. D. & Feldmann, J. Laser printing single gold nanoparticles. *Nano Lett.* **10**, 4794-4798 (2010).
- 14 Whitesides, G. M. & Grzybowski, B. Self-assembly at all scales. *Science* **295**, 2418-2421 (2002).
- 15 Shi, J. et al. Modular assembly of optical nanocircuits. *Nat. Commun.* **5**, 3896 (2014).
- 16 Lin, L. H. et al. All-optical reconfigurable chiral meta-molecules. *Mater. Today* **25**, 10-20 (2019).
- 17 Kuzyk, A. et al. Selective control of reconfigurable chiral plasmonic metamolecules. *Sci. Adv.* **3**, e1602803 (2017).

- 18 Lal, S. et al. Noble metal nanowires: From plasmon waveguides to passive and active devices. *Acc. Chem. Res.* **45**, 1887-1895 (2012).
- 19 Guo, X. et al. Direct coupling of plasmonic and photonic nanowires for hybrid nanophotonic components and circuits. *Nano Lett.* **9**, 4515-4519 (2009).

Reviewers' comments:

Reviewer #1 (Remarks to the Author):

The authors have done a good job of addressing all the comments from the previous reviewers. It would be good to include an abridged version of the response to Reviewer 2's first comment, about the differences between the work in this paper and the authors' 2018 Nature Photonics paper.

Reviewer #2 (Remarks to the Author):

Whilst some of the answers in reply are clear, it still seems ambiguous about where this approach is used vs the Nature Photonics work of the authors. In particular the authors are unclear about area of applicability. In the abstract they infer this method can be used in " liquid, vapor, or vacuum" but this is not clear in the manuscript

I also think the phrases regarding self -organisation are still misleading without real data showing organisation of large numbers (eg 50-100) particles

Reviewer #3 (Remarks to the Author):

The authors have addressed most weak points I raised in my prior review.
I still have some concerns on the position accuracy that can be achieved by their technique.

In my previous review, I asked " the averaged particle position deviation is ~200 nm. How can this resolution allows the control of assembled nanoscale particles. For example, can this technique be reliably used to control the 15nm gap spacing of a dimer of AuNP?" The authors addressed this issue by proposing that the gap spacing between two Au dimers can be accurately controlled by monitoring the spectrum shift that is correlated to the gap spacing. Although this is true, this approach, however, also introduces large variation when manipulating different shapes of Au nanoparticles, different kinds of nanoparticles, and different kinds of substrate since these factors all affect the scattering spectrum one will detect.

Since this manuscript emphasizes its ability on manipulating nanoparticles, I wish to see claims to be made more precisely to reflect the true capability and limitations of this tool, especially on nanoparticle patterning.

I also went through authors' answers to other reviewers' questions. I think the authors address reviewer 2's questions well. For reviewer 1's question, there are parts such as MD simulation and resistance force that I cannot provide good adjustment on the accuracy of these answers based on my limited knowledge.

Reviewer #1:

The authors have done a good job of addressing all the comments from the previous reviewers. It would be good to include an abridged version of the response to Reviewer 2's first comment, about the differences between the work in this paper and the authors' 2018 Nature Photonics paper.

Answer: Thank you very much for the reviewer's positive comments and further suggestions. As stated in the previous response letter, this work is different from our recent 2018 Nature Photonics paper in terms of the phase of CTAC, the function of CTAC, working environment, driving force, and the role of optical heating. In the revised manuscript, we have included a brief discussion to show the main differences of these two works as suggested by the reviewer.

Page 5, Lines 83-87, "It should be noted that this work is quite different from opto-thermoelectric nanotweezers published in a recent paper¹. In our OPN, particles are manipulated on solid substrates by optical scattering forces. In contrast, opto-thermoelectric nanotweezers exploit CTAC that is dissolved into a colloidal solution to generate an opto-thermoelectric field to trap charged nanoparticles." was added.

Reviewer #2:

Whilst some of the answers in reply are clear, it still seems ambiguous about where this approach is used vs the Nature Photonics work of the authors.

Answer: Thank you for the reviewer's comments. In this work, we manipulate particles on a *solid* substrate using optical *scattering* forces. CTAC is used in its solid phase and optical heating is exploited to realize the localized phase transition of CTAC from solid to quasi-liquid phase. *Particles maintain on substrate after the laser is turn off*. While in the Nature Photonics paper¹, nanoparticles are manipulated in *liquid* environment by *opto-thermoelectric* forces. CTAC is added into solution to form CTA⁺

micelles and Cl^- ions and optical heating is employed to generate a temperature gradient in solution to induce the thermophoretic migration of CTA^+ micelles and Cl^- ions. *Particles are released after the laser is off.* The different working mechanisms and conditions make these two approaches suitable for the different applications. We have revised the manuscript to clarify the main differences of these two works in both working mechanism and applicability (see response to Reviewer #1).

In particular the authors are unclear about area of applicability. In the abstract they infer this method can be used in " liquid, vapor, or vacuum" but this is not clear in the manuscript.

Answer: Thank you for the reviewer's comments. Conventional optical manipulations have been used for manipulation of colloidal particles in liquid, vapor, or vacuum environment. In this work, we develop the solid-phase optical nanomanipulation to extend the capabilities of optical manipulations to achieve the manipulation of colloidal particles on solid substrates directly. Thus, we previously claimed that our technique "breaks the limitations of conventional optical tweezers to liquid, vapor, or vacuum environments" and can be used for manipulation of colloidal particles on a solid substrate. We didn't indicate that this technique can be used in any "liquid, vapor, or vacuum". However, it is worth noting that our technique *is compatible with* any solid substrate, vacuum, any vapor or liquid solution as long as the optothermal gate layer is not dissolved or damaged. To avoid any unclear claims, we deleted the related sentences in the Abstract in the revised manuscript.

Page 2, Line 18, "breaking the limitations of conventional optical tweezers to liquid, vapor, or vacuum environments." *was deleted.*

I also think the phrases regarding self-organisation are still misleading without real data showing organisation of large numbers (eg 50-100) particles.

Answer: Thank you for the reviewer’s comments. In this manuscript, we demonstrated “light-directed patterning” of colloidal particles, not “self-organisation”. Also, we didn’t mention any phrases regarding self-organisation throughout our manuscript and supplementary information. Although self-organisation is well-known for large-scale, automatic formation of colloidal structures, our OPN also creates complex colloidal nanostructures with nanoparticles easily. For instance, in the revised manuscript, we demonstrated the patterning of a triangular spiral pattern composed of 9 AuNPs and a “Au” pattern composed of 22 AuNPs (Fig. R1, also see Supplementary Fig. 10). Another advantage of our technique over self-assembly is that the patterns achieved by OPN are not limited to entropically stable configurations.

Pages 11-12, Lines 206-207, “Moreover, OPN is capable of on-demand patterning of colloidal particles into more complex configurations (Supplementary Fig. 10).” was added.

Figure R1. Patterning of 80 nm AuNPs into (a) a triangular spiral composed of 9 particles and (b) a “Au” pattern composed of 22 particles. Scale bars: a, 2 μm ; b, 5 μm .

Reviewer #3:

The authors have addressed most weak points I raised in my prior review.

I still have some concerns on the position accuracy that can be achieved by their technique.

Answer: Thank you so much for the reviewer's comments. We have made point-to-point responses to the further concerns raised by the reviewer:

In my previous review, I asked "the averaged particle position deviation is ~200 nm. How can this resolution allows the control of assembled nanoscale particles. For example, can this technique be reliably used to control the 15nm gap spacing of a dimer of AuNP?" The authors addressed this issue by proposing that the gap spacing between two Au dimers can be accurately controlled by monitoring the spectrum shift that is correlated to the gap spacing. Although this is true, this approach, however, also introduces large variation when manipulating different shapes of Au nanoparticles, different kinds of nanoparticles, and different kinds of substrate since these factors all affect the scattering spectrum one will detect.

Answer: Thank you for the reviewer's comments. In the manuscript, we proposed that the gap spacing between two colloidal particles can be accurately controlled by monitoring the scattering spectrum of the dimer that is correlated with the gap spacing; we also experimentally demonstrate the reliable fabrication of Au nanosphere dimers with controllable gaps. We agree with the reviewer's comments that the scattering spectra are affected by different kinds of nanoparticles, different kinds of substrates, and different shapes of nanoparticles. However, this proposed strategy is still applicable for different kinds of nanoparticles and substrates as long as the scattering spectrum is solely dependent on the interparticle gaps in each configuration. For anisotropic nanoparticles (*e.g.*, Au nanotriangles, Au nanocubes, Au nanostars, Au nanobipyramids, and Au nanorods), the configurations of particle assembly and the corresponding scattering spectra are largely determined by the orientation of these nanoparticles. Since it is difficult to control the orientation of anisotropic nanoparticles due to the diffraction limit, it remains

challenging to measure the interparticle gaps using this method in this case. Nevertheless, in certain cases, such as Au nanorods, the orientation of the scattering spectrum can be detected by polarization-dependent measurement. Therefore, it is still possible to detect the interparticle gap by studying the polarization-resolved scattering spectra. Despite of this limitation, our technique can *in situ* monitor the scattering spectra of colloidal structures and estimate the interparticle gaps during the assembly process, which is favourable in the investigation of light-matter interactions and coupling of colloidal particles. In the revised manuscript, we mentioned the area of applicability of this strategy, highlighted the advantages, and discussed the limitations of this strategy in detecting the gap of anisotropic nanoparticles.

Supplementary Information, Supplementary Note 6, Pages 18-19, Lines 248-253:

“This strategy is generally applicable to all types of nanoparticles and substrates. However, the effects of orientations of anisotropic nanoparticles need be considered in establishing the relationship between the interparticle gaps and scattering spectra. The *in situ* optical spectroscopy that can monitor the scattering spectra of colloidal structures and estimate the interparticle gaps during the assembly process will significantly benefit the investigation of light-matter interactions and coupling of colloidal particles.”
was added.

Since this manuscript emphasizes its ability on manipulating nanoparticles, I wish to see claims to be made more precisely to reflect the true capability and limitations of this tool, especially on nanoparticle patterning.

Answer: Thank you for the reviewer’s comments. Our technique can versatily manipulate a wide range of colloidal particles that interact with light on a solid substrate. Colloidal particles can be translated to any targeted positions and patterned into any desired configurations with nanoscale

accuracy. There are two limitations. First, as an optical technique, the imaging resolution is limited by the diffraction limit. Thus, sub-20 nm patterning accuracy and orientational control of anisotropic nanoparticles remain challenging. Second, the current patterning throughput is relatively low since we rely on the manual control to manipulate colloidal particles. This can be potentially improved by the implementation of a light spatial modulator with a digital feedback control. We have discussed the low-throughput and the proposed possible solution in the Discussion (Page 19, Lines 324-328). In the revised manuscript, we further added discussion on the limitations in patterning accuracy and orientational control of anisotropic nanoparticles.

Page 18, Lines 319-322, “OPN can dynamically pattern colloidal particles into any desired configurations. However, it remains challenging to achieve sub-20 nm position accuracy and orientational control of anisotropic nanoparticles due to the optical diffraction limit.” **was added.**

I also went through authors' answers to other reviewers' questions. I think the authors address reviewer 2's questions well. For reviewer 1's question, there are parts such as MD simulation and resistance force that I cannot provide good adjustment on the accuracy of these answers based on my limited knowledge.

Answer: Thank you for the reviewer's comments. Molecular dynamics simulations have been widely used in the calculation of the viscosity of organic liquid since decades ago². It is a reliable and mature method to predict viscosity. Simulations match well with experiments especially in the range of 0.1-10 cP^{2,3}. In our case, the viscosity of CTAC at ~450 K is in this range. Thus, the viscosity calculation of CTAC should be accurate. The resistant force of moving particles partially immersed in liquid has also been extensively studied^{4,7}. The modified Stokes equation used in this manuscript has been widely studied by both theory⁵ and experiments^{4,7}. The model used here can give a relatively accurate estimation of resistant forces of colloidal particles during the particle manipulation process.

Other changes made:

The Abstract was shortened to meet the length requirement for *Nature Communications*:

Page 2, Line 17, “between particles and a substrate” **was deleted.**

Page 2, Line 18, “by optically heating the particles with a laser beam” **was replaced by** “by optically heating the particles”.

Page 2, Lines 21-22, “OPN is capable of on-demand manipulation and patterning of colloidal particles of variable sizes (from sub-50 nm to micrometre scale), shapes (e.g., spheres and wires) and compositions (e.g., gold, silver, silicon, titanium dioxide, and metal-dielectric hybrids) at nanoscale accuracy” **was replaced by** “OPN is capable of on-demand manipulation and patterning of colloidal particles of variable sizes, shapes, and compositions at nanoscale accuracy”.

Page 2, Line 22, “Along with in situ optical spectroscopy for characterizations of the colloidal particles” **was replaced by** “Along with in situ optical spectroscopy”.

Page 2, Line 23, “solid-phase” **was deleted.**

References

- 1 Lin, L. et al. Opto-thermoelectric nanotweezers. *Nat. Photonics* **12**, 195-201 (2018).
- 2 Mondello, M. & Grest, G. S. Viscosity calculations of n-alkanes by equilibrium molecular dynamics. *J. Chem. Phys.* **106**, 9327-9336 (1997).
- 3 Zhang, Y., Otani, A. & Maginn, E. J. Reliable viscosity calculation from equilibrium molecular dynamics simulations: A time decomposition method. *J. Chem. Theory Comput.* **11**, 3537-3546 (2015).
- 4 Petkov, J. T. et al. Measurement of the drag coefficient of spherical particles attached to fluid interfaces. *J. Colloid. Interface Sci.* **172**, 147-154 (1995).
- 5 Danov, K., Aust, R., Durst, F. & Lange, U. Influence of the surface viscosity on the hydrodynamic resistance and surface diffusivity of a large brownian particle. *J. Colloid. Interface Sci.* **175**, 36-45 (1995).
- 6 Kralchevsky, P. A. & Nagayama, K. Capillary interactions between particles bound to interfaces, liquid films and biomembranes. *Adv. Colloid. Interfaces* **85**, 145-192 (2000).
- 7 Petkov, J. T. et al. Precise method for measuring the shear surface viscosity of surfactant monolayers. *Langmuir* **12**, 2650-2653 (1996).

REVIEWERS' COMMENTS:

Reviewer #3 (Remarks to the Author):

The authors have addressed all my concerns. I have no further questions.

Reviewer #3:

The authors have addressed all my concerns. I have no further questions.

Answer: Thank you so much for the reviewer's comments. We thank all the reviewers for their constructive suggestions and positive comments during the review process.